# THE RICH GET RICHER:
# DISPARATE IMPACT OF SEMI-SUPERVISED LEARNING

**Zhaowei Zhu**[*]**, Tianyi Luo**[*]**, and Yang Liu**
Computer Science and Engineering, University of California, Santa Cruz
`{zwzhu,tluo6,yangliu}@ucsc.edu`

## ABSTRACT

Semi-supervised learning (SSL) has demonstrated its potential to improve the model accuracy for a variety of learning tasks when the high-quality supervised data is severely limited. Although it is often established that the average accuracy for the entire population of data is improved, it is unclear how SSL fares with different sub-populations. Understanding the above question has substantial fairness implications when different sub-populations are defined by the demographic groups that we aim to treat fairly. In this paper, we reveal the disparate impacts of deploying SSL: the sub-population who has a higher baseline accuracy without using SSL (the "rich" one) tends to benefit more from SSL; while the sub-population who suffers from a low baseline accuracy (the "poor" one) might even observe a performance drop after adding the SSL module. We theoretically and empirically establish the above observation for a broad family of SSL algorithms, which either explicitly or implicitly use an auxiliary "pseudo-label". Experiments on a set of image and text classification tasks confirm our claims. We introduce a new metric, *Benefit Ratio*, and promote the evaluation of the fairness of SSL (Equalized Benefit Ratio). We further discuss how the disparate impact can be mitigated. We hope our paper will alarm the potential pitfall of using SSL and encourage a multifaceted evaluation of future SSL algorithms.

## 1 INTRODUCTION

The success of deep neural networks benefits from large-size datasets with high-quality supervisions, while the collection of them may be difficult due to the high cost of data labeling process (Agarwal et al., 2016; Wei et al., 2022). A much cheaper and easier solution is to obtain a small labeled dataset and a large unlabeled dataset, and apply semi-supervised learning (SSL). Although the global model performance for the entire population of data is almost always improved by SSL, it is unclear how the improvements fare for different sub-populations, e.g., a multiaccuracy metric (Kim et al., 2019). Analyzing and understanding the above question will have substantial fairness implications especially when sub-populations are defined by the demographic groups e.g., race and gender.

**Motivating Example** Figure 1 shows the change of test accuracy for each class during SSL. Each class denotes one sub-population and representative sub-populations are highlighted. Figure 1 delivers two important messages: in SSL, even with some state-of-the-art algorithms: 1) the observation "rich getting richer" is common, and 2) the observation "poor getting poorer" possibly happens. Specifically, the *"rich"* sub-population, such as *automobile* that has a high baseline accuracy at the beginning of SSL, tends to be consistently benefiting from SSL. But the *"poor"* sub-population, such as *dog* that has a low baseline accuracy, will remain a low-level performance as Figure 1(a) or even get worse performance as Figure 1(b). This example shows disparate impact of accuracies for different sub-populations are common in SSL. In special cases such as Figure 1(b), we observe the Matthew effect: the rich get richer and the poor get poorer. See more discussions in Appendix B.

In this paper, we aim to understand the disparate impact of SSL from both theoretical and empirical aspects, and propose to evaluate SSL from a different dimension. Specifically, based on classifications tasks, we study the disparate impact of model accuracies with respect to different sub-populations (such as label classes, feature groups and demographic groups) after applying SSL.

---

[*]Equal contributions. Corresponding author: Yang Liu <`yangliu@ucsc.edu`>.

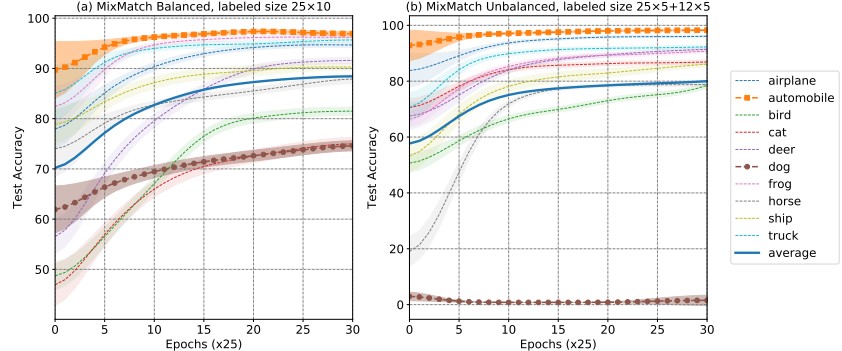

Figure 1: Disparate impacts in the model accuracy of SSL. MixMatch (Berthelot et al., 2019b) is applied on CIFAR-10 with (a) 250 clean labels in the balanced case (25 labeled instances per class) and (b) 185 clean labels in the unbalanced case (25 labeled instances in each of the first 5 classes, 12 labeled instances in each of the remaining classes). Other instances are used as unlabeled data.

Different from traditional group fairness (Zafar et al., 2017; Bellamy et al., 2018; Jiang et al., 2022b) defined over one model, our study focuses on comparing the gap between two models (before and after SSL). To this end, we first theoretically analyze why and how disparate impact are generated. The theoretical results motivate us to further propose a new metric, *benefit ratio*, which evaluates the normalized improvement of model accuracy using SSL methods for different sub-populations. The benefit ratio helps reveal the *"Matthew effect"* of SSL: a high baseline accuracy tends to reach a high benefit ratio that may even be larger than 1 (the rich get richer), and a sufficiently low baseline accuracy may return a negative benefit ratio (the poor get poorer). The above revealed Matthew effect indicates that existing and popular SSL algorithms can be unfair. Aligning with recent literature on fair machine learning (Hardt et al., 2016; Corbett-Davies & Goel, 2018; Verma & Rubin, 2018), we promote that a fair SSL algorithm should benefit different sub-populations equally, i.e., achieving Equalized Benefit Ratio which we will formally define in Definition 1. We then evaluate SSL using benefit ratios and discuss how two possible treatments, i.e., balancing the data and collecting more labeled data, might mitigate the disparate impact. We hope our analyses and discussions could encourage future contributions to promote the fairness of SSL.

Our main contributions and findings are summarized as follows: 1) We propose an analytical framework that unifies a broad family of SSL algorithms, which either explicitly or implicitly use an auxiliary "pseudo-label". The framework provides a novel perspective for analyzing SSL by connecting consistency regularization to learning with noisy labels. 2) Based on our framework, we further prove an upper bound for the generalization error of SSL and theoretically show the heterogeneous error of learning with the smaller scale supervised dataset is one primary reason for disparate impacts. This observation reveals a "Matthew effect" of SSL. 3) We contribute a novel metric called *benefit ratio* to measure the disparate impact in SSL, the effectiveness of which is guaranteed by our proved generalization bounds. 4) We conduct experiments on both image and text classification tasks to demonstrate the ubiquitous disparate impacts in SSL. 5) We also discuss how the disparate impact could be mitigated. Code is available at `github.com/UCSC-REAL/Disparate-SSL`.

## 1.1 RELATED WORKS

**Semi-supervised learning**  SSL is popular in various communities (Dai & Le, 2015; Howard & Ruder, 2018; Clark et al., 2018; Yang et al., 2019; Sachan et al., 2019; Guo et al., 2020; Cheng et al., 2021; Wang et al., 2021c; Luo et al., 2021; Huang et al., 2021; Bai et al., 2021; Wang et al., 2020; Liu et al., 2020). We briefly review recent advances in SSL. See comprehensive overviews by (Chapelle et al., 2006; Zhu et al., 2003) for traditional methods. Recent works focus on assigning pseudo-labels generated by the supervised model to unlabeled dataset (Lee et al., 2013; Iscen et al., 2019; Berthelot et al., 2019b;a), where the pseudo-labels are often confident or with low-entropy (Sohn et al., 2020; Zhou, 2018; Meng et al., 2018). There are also many works on minimizing entropy of predictions on unsupervised data (Grandvalet et al., 2005) or regularizing the model consistency on the same feature with different data augmentations (Tarvainen & Valpola, 2017; Miyato et al., 2018; Sajjadi et al., 2016; Zhang et al., 2018; Miyato et al., 2016; Xie et al., 2019). In addition to network inputs, augmentations can also be applied on hidden layers (Chen et al., 2020). Besides, some works (Peters et al., 2018; Devlin et al., 2018; Gururangan et al., 2019; Chen et al., 2019; Yang et al., 2017)

first conduct pre-training on the unlabeled dataset then fine-tune on the labeled dataset, or use ladder networks to combine unsupervised learning with supervised learning (Rasmus et al., 2015).

**Disparate impact** Even models developed with the best intentions may introduce discriminatory biases (Raab & Liu, 2021). Researchers in various fields have found the unfairness issues, e.g., vision-and-language representations (Wang et al., 2022), model compression (Bagdasaryan et al., 2019), differential privacy (Hooker et al., 2019; 2020), recommendation system (Gómez et al., 2021), information retrieval (Gao & Shah, 2021), image search (Wang et al., 2021b), machine translation (Khan et al., 2021), message-passing (Jiang et al., 2022a), and learning with noisy labels (Liu, 2021; Zhu et al., 2021a; Liu & Wang, 2021). There are also some treatments considering fairness without demographics (Lahoti et al., 2020; Diana et al., 2021; Hashimoto et al., 2018), minimax Pareto fairness (Martinez et al., 2020), multiaccuracy boosting (Kim et al., 2019), and fair classification with label noise (Wang et al., 2021a). Most of these works focus on supervised learning. To our best knowledge, the unfairness of SSL has not been sufficiently explored.

## 2 PRELIMINARIES

We summarize the key concepts and notations as follows.

### 2.1 SUPERVISED CLASSIFICATION TASKS

Consider a $K$-class classification task given a set of $N_L$ labeled training examples denoted by $D_L := \{(x_n, y_n)\}_{n \in [N_L]}$, where $[N_L] := \{1, 2, \cdots, N_L\}$, $x_n$ is an input feature, $y_n \in [K]$ is a label. The clean data distribution with full supervision is denoted by $\mathcal{D}$. Examples $(x_n, y_n)$ are drawn according to random variables $(X, Y) \sim \mathcal{D}$. The classification task aims to identify a classifier $f$ that maps $X$ to $Y$ accurately. Let $\mathbb{1}\{\cdot\}$ be the indicator function taking value 1 when the specified condition is satisfied and 0 otherwise. Define the *0-1 loss* as $\mathbb{1}(f(X), Y) := \mathbb{1}\{f(X) \neq Y\}$. The optimal $f$ is denoted by the Bayes classifier $f^* = \arg\min_f \mathbb{E}_{\mathcal{D}}[\mathbb{1}(f(X), Y)]$. One common choice is training a deep neural network (DNN) by minimizing the empirical risk: $\hat{f} = \arg\min_f \frac{1}{N} \sum_{n=1}^{N} \ell(f(x_n), y_n)$. Notation $\ell(\cdot)$ stands for the cross-entropy (CE) loss $\ell(f(x), y) := -\ln(\boldsymbol{f}_x[y]), y \in [K]$, where $\boldsymbol{f}_x[y]$ denotes the $y$-th component of $\boldsymbol{f}(x)$. Notations $f$ and $\boldsymbol{f}$ stand for the same model but different outputs. Specifically, vector $\boldsymbol{f}(x)$ denotes the probability of each class that model $f$ predicts given feature $x$. The predicted label $f(x)$ is the class with maximal probability, i.e., $f(x) := \arg\max_{i \in [K]} \boldsymbol{f}_x[i]$. We use notation $f$ if we only refer to a model.

### 2.2 SEMI-SUPERVISED CLASSIFICATION TASKS

In the semi-supervised learning (SSL) task, there is also an unlabeled (a.k.a. unsupervised) dataset $D_U := \{(x_{n+N_L}, \cdot)\}_{n \in [N_U]}$ drawn from $\mathcal{D}$, while the labels are missing or unobservable. Let $N := N_L + N_U$. Denote the corresponding unobservable supervised dataset by $D := \{(x_n, y_n)\}_{n \in [N]}$. Compared with the supervised learning tasks, it is critical to leverage the unsupervised information in semi-supervised learning. To improve the model generalization ability, many recent SSL methods build consistency regularization with unlabeled data to ensure that the model output remains unchanged with randomly augmented inputs (Berthelot et al., 2019b; Xie et al., 2019; Berthelot et al., 2019a; Sohn et al., 2020; Xu et al., 2021). To proceed, we introduce soft/pseudo-labels $\boldsymbol{y}$.

**Soft-labels** Note the one-hot encoding of $y_n$ can be written as $\boldsymbol{y}_n$, where each element writes as $\boldsymbol{y}_n[i] = \mathbb{1}\{i = y_n\}$. More generally, we can extend the one-hot encoding to soft labels by requiring each element $\boldsymbol{y}[i] \in [0, 1]$ and $\sum_{i \in [K]} \boldsymbol{y}[i] = 1$. The CE loss with soft $\boldsymbol{y}$ writes as $\ell(\boldsymbol{f}(x), \boldsymbol{y}) := -\sum_{i \in [K]} \boldsymbol{y}[i] \ln(\boldsymbol{f}_x[i])$. If we interpret $\boldsymbol{y}[i] = \mathbb{P}(Y = i)$ as a probability (Zhu et al., 2022a) and denote by $\mathcal{D}_{\boldsymbol{y}}$ the corresponding label distribution, the above CE loss with soft labels can be interpreted as the expected loss with respect to a stochastic label $\widetilde{Y}$, i.e.,

$$\ell(\boldsymbol{f}(x), \boldsymbol{y}) := \sum_{i \in [K]} \mathbb{P}(\widetilde{Y} = i)\ell(f(x), i) = \mathbb{E}_{\widetilde{Y} \sim \mathcal{D}_{\boldsymbol{y}}}\left[\ell(f(x), \widetilde{Y})\right]. \tag{1}$$

**Pseudo-labels** In consistency regularization, by using model predictions, the unlabeled data will be assigned pseudo-labels either explicitly (Berthelot et al., 2019b) or implicitly (Xie et al., 2019), where the pseudo-labels can be modeled as soft-labels. In the following, we review both the explicit and the implicit approaches and unify them in our analytical framework.

**Consistency regularization with explicit pseudo-labels** For each unlabeled feature $x_n$, pseudo-labels can be explicitly generated based on model predictions (Berthelot et al., 2019b). The pseudo-label is later used to evaluate the model predictions. To avoid trivial solutions where model predictions and pseudo-labels are always identical, independent data augmentations of feature $x_n$ are often generated for $M$ rounds. The augmented feature is denoted by $x'_{n,m} := \text{Augment}(x_n), m \in [M]$. Then the pseudo-label $\boldsymbol{y}_n$ in epoch-$t$ can be determined based on $M$ model predictions as $\boldsymbol{y}_n^{(t)} = \text{Sharpen}\left(\frac{1}{M}\sum_{m=1}^{M} \bar{\boldsymbol{f}}^{(t)}(x'_{n,m})\right)$, where model $\bar{f}^{(t)}$ is a copy of the DNN at the beginning of epoch-$t$ but without gradients. The function $\text{Sharpen}(\cdot)$ reduces the entropy of a pseudo-label, e.g., setting to one-hot encoding $\boldsymbol{e}_j, j = \arg\max_{i\in[K]} \boldsymbol{y}_n^{(t)}$ (Sohn et al., 2020). In epoch-$t$, with some consistency regularization loss $\ell_{\text{CR}}(\cdot)$, the empirical risk using pseudo-labels is:

$$L_1(f, D_L, D_U) = \frac{1}{N_L}\sum_{n=1}^{N_L} \ell(f(x_n), y_n) + \frac{1}{N_U}\sum_{n=N_L+1}^{N_L+N_U} \ell_{\text{CR}}(\boldsymbol{f}(x_n), \boldsymbol{y}_n^{(t)}).$$

**Consistency regularization with implicit pseudo-labels** Consistency regularization can also be applied without specifying particular pseudo-labels, where a divergence metric between predictions on the original feature and the augmented feature is minimized to make predictions consistent. For example, the KL-divergence could be applied and the data augmentation could be domain-specific (Xie et al., 2019) or adversarial (Miyato et al., 2018). In epoch-$t$, the total loss is:

$$L_2(f, D_L, D_U) = \frac{1}{N_L}\sum_{n=1}^{N_L} \ell(f(x_n), y_n) + \lambda \cdot \frac{1}{N_U}\sum_{n=N_L+1}^{N_L+N_U} \ell_{\text{CR}}(\boldsymbol{f}(x_n), \bar{\boldsymbol{f}}^{(t)}(x'_n)),$$

where $\lambda$ balances the supervised loss and the unsupervised loss, $x'_n := \text{Augment}(x_n)$ stands for one-round data augmentation ($m = 1$ following the previous notation $x'_{n,m}$). Without loss of generality, we use $\lambda = 1$ in our analytical framework.

**Consistency regularization loss $\ell_{\text{CR}}(\cdot)$** In the above two lines of works, there are different choices of $\ell_{\text{CR}}(\cdot)$, such as mean-squared error $\ell_{\text{CR}}(\boldsymbol{f}(x), \boldsymbol{y}) := \|\boldsymbol{f}(x) - \boldsymbol{y}\|_2^2/K$ (Berthelot et al., 2019b; Laine & Aila, 2016) or CE loss (Miyato et al., 2018; Xie et al., 2019) defined in Eq. (1). For a clean analytical framework, we consider the case when both supervised loss and unsupervised loss are the same, i.e., $\ell_{\text{CR}}(\boldsymbol{f}(x), \boldsymbol{y}) = \ell(\boldsymbol{f}(x), \boldsymbol{y})$. Note $L_2$ implies the entropy minimization (Grandvalet et al., 2005) when both loss functions are CE and there is no augmentation, i.e., $x'_n = x_n$.

## 2.3 ANALYTICAL FRAMEWORK

We propose an analytical framework to unify both the explicit and the implicit approaches. With Eqn. (1), the unsupervised loss in the above two methods can be respectively written as:

$$\frac{1}{N_U}\sum_{n=N_L+1}^{N_L+N_U} \mathbb{E}_{\widetilde{Y}\sim\mathcal{D}_{\boldsymbol{y}_n^{(t)}}}\left[\ell(f(x_n), \widetilde{Y})\right] \quad \text{and} \quad \frac{1}{N_U}\sum_{n=N_L+1}^{N_L+N_U} \mathbb{E}_{\widetilde{Y}\sim\mathcal{D}_{\bar{f}^{(t)}(x'_n)}}\left[\ell(f(x_n), \widetilde{Y})\right].$$

Both unsupervised loss terms inform us that, for each feature $x_n$, we compute the loss with respect to the reference label $\widetilde{Y}$, which is a random variable following distribution $\mathcal{D}_{\boldsymbol{y}_n^{(t)}}$ or $\mathcal{D}_{\bar{f}^{(t)}(x'_n)}$. Compared with the original clean label $Y$, the unsupervised reference label $\widetilde{Y}$ contains label noise, and the noise transition (Zhu et al., 2021b) depends on feature $x_n$. Specifically, we have

$$\mathbb{P}(\widetilde{Y} = i|X = x_n) = \boldsymbol{y}_n^{(t)}[i] \text{ (Explicit)} \quad \text{or} \quad \mathbb{P}(\widetilde{Y} = i|X = x_n) = \bar{f}_{x'_n}^{(t)}[i] \text{ (Implicit)}.$$

Then with model $\bar{f}^{(t)}$, we can define a new dataset with instance-dependent noisy reference labels: $\widetilde{D} = \{(x_n, \tilde{\boldsymbol{y}}_n)\}_{n\in[N]}$, where $\tilde{\boldsymbol{y}}_n = \boldsymbol{y}_n, \forall n \in [N_L]$, and $\tilde{\boldsymbol{y}}_n = \boldsymbol{y}_n^{(t)}$ or $\bar{f}_{x'_n}^{(t)}, \forall n \in \{N_L+1, \cdots, N\}$. The unified loss is:

$$L(f, \widetilde{D}) = \frac{1}{N}\sum_{n=1}^{N} \ell(\boldsymbol{f}(x_n), \tilde{\boldsymbol{y}}_n) = \frac{1}{N}\sum_{n=1}^{N} \mathbb{E}_{\widetilde{Y}\sim\mathcal{D}_{\tilde{\boldsymbol{y}}_n}}\left[\ell(\boldsymbol{f}(x_n), \widetilde{Y})\right]. \tag{2}$$

Therefore, with the pseudo-label as a bridge, we can model the popular SSL solution with consistency regularization as a problem of learning with noisy labels (Natarajan et al., 2013). But different from the traditional class-dependent assumption (Liu & Tao, 2015; Wei & Liu, 2021; Zhu et al., 2022b), the instance-dependent pseudo-labels are more challenging (Liu, 2022).

## 3 DISPARATE IMPACTS OF SSL

To understand the disparate impact of SSL, we study how supervised learning with $D_L$ affects the performance of SSL with both $D_L$ and $D_U$. In this section, the analyses are for one and an arbitrary sub-population, thus we did not explicitly distinguish sub-populations in notation.

**Intuition** The rich sub-population with a higher baseline accuracy (from supervised learning with $D_L$) will have higher-quality pseudo-labels for consistency regularization, which helps to further improve the performance. In contrast, the poorer sub-population with a lower baseline accuracy (again from supervised learning with $D_L$) can only have lower-quality pseudo-labels to regularize the consistency of unsupervised features. With a wrong regularization direction, the unsupervised feature will have its augmented copies reach consensus on a wrong label class, which leads to a performance drop. Therefore, when the baseline accuracy is getting worse, there will be more and more unsupervised features that are wrongly regularized, resulting in disparate impacts of model accuracies as shown in Figure 1.

In the following, we first analyze the generalization error for supervised learning with labeled data, then extend the analyses to semi-supervised learning with the help of pseudo-labels. Without loss of generality, we consider minimizing 0-1 loss $\mathbb{1}(f(X), Y)$ with infinite search space. Our analyses can be generalized to bounded loss $\ell(\cdot)$ and finite function space $\mathcal{F}$ following the generalization bounds that can be introduced using Rademacher complexity (Bartlett & Mendelson, 2002).

### 3.1 LEARNING WITH CLEAN DATA

Denote the expected error rate of classifier $f$ on distribution $\mathcal{D}$ by $R_{\mathcal{D}}(f) := \mathbb{E}_{\mathcal{D}}[\mathbb{1}(f(X), Y)]$. Let $\hat{f}_D$ denote the classifier trained by minimizing 0-1 loss with clean dataset $D$, i.e., $\hat{f}_D := \arg\min_f \hat{R}_D(f)$, where $\hat{R}_D(f) := \frac{1}{N}\sum_{n\in[N]}\mathbb{1}(f(x_n), y_n)$. Denote by $Y^*|X := \arg\max_{i\in[K]}\mathbb{P}(Y|X)$ the Bayes optimal label on clean distribution $\mathcal{D}$. Theorem 1 shows the generalization bound in the clean case. Replacing $D$ and $N$ with $D_L$ and $N_L$ we have:

**Theorem 1** (Supervised error). *With probability at least $1-\delta$, the generalization error of supervised learning on clean dataset $D_L$ is upper-bounded by $R_{\mathcal{D}}(\hat{f}_{D_L}) \leq \sqrt{\frac{2\log(2/\delta)}{N_L}} + \mathbb{P}(Y^* \neq Y)$.*

### 3.2 LEARNING WITH SEMI-SUPERVISED DATA

We further derive generalization bounds for learning with semi-supervised data. Following our analytical framework in Section 2.3, in each epoch-$t$, we can transform the semi-supervised data to supervised data with noisy supervisions by assigning pseudo-labels, where the semi-supervised dataset $D_L \cup D_U$ is converted to the dataset $\widetilde{D}$ given the model learned from the previous epoch.

**Two-iteration scenario** In our theoretical analyses, to find a clean-structured performance bound for learning with semi-supervised data and highlight the impact of the model learned from supervised data, we consider a particular *two-iteration scenario* where the model is first trained to convergence with the small labeled dataset $D_L$ and get $\hat{f}_{D_L}$, then trained on the pseudo noisy dataset $\widetilde{D}$ labeled by $\hat{f}_{D_L}$. Noting assigning pseudo labels iteratively may improve the performance as suggested by most SSL algorithms (Berthelot et al., 2019b; Xie et al., 2019), our considered two-iteration scenario can be seen as a worst case for an SSL algorithm.

**Independence of samples in $\widetilde{D}$** Recall $x'_n$ denotes the augmented copy of $x_n$. The $N$ instances in $\widetilde{D}$ may not be independent since pseudo-label $\widetilde{\boldsymbol{y}}_n$ comes from $\hat{f}_{D_L}(x'_n)$, which depends on $x_n$. Namely, the number of independent instances $N'$ should be in the range of $[N_L, N]$. Intuitively, with appropriate noise injection or data augmentation (Xie et al., 2019; Miyato et al., 2018) to $x_n$ such that $x'_n$ could be treated as independent of $x_n$, the number of independent samples in $\widetilde{D}$ could be improved to $N$. We consider the ideal case where all $N$ instances are *i.i.d.* in the following analyses.

By minimizing the unified loss defined in Eq. (2), we can get classifier $\hat{f}_{\widetilde{D}} := \arg\min_f \hat{R}_{\widetilde{D}}(f)$, where $\hat{R}_{\widetilde{D}}(f) := \frac{1}{N}\sum_{n\in[N]}\left(\sum_{i\in[K]}\tilde{\boldsymbol{y}}[i] \cdot \mathbb{1}(f(x_n), i)\right)$. The expected error given classifier $f$ is

denoted by $R_{\widetilde{\mathcal{D}}}(f) := \mathbb{E}_{\widetilde{\mathcal{D}}}[\mathbb{1}(f(X), \widetilde{Y})]$, where the probability density function of distribution $\widetilde{\mathcal{D}}$ can be defined as $\mathbb{P}_{(X,\widetilde{Y})\sim\widetilde{\mathcal{D}}}(X = x_n, \widetilde{Y} = i) = \mathbb{P}_{(X,Y)\sim\mathcal{D}}(X = x_n) \cdot \tilde{\boldsymbol{y}}_n[i]$.

**Decomposition**  With the above definitions, the generalization error (on the clean distribution) of classifier $\hat{f}_{\widetilde{D}}$ could be decomposed as $R_{\mathcal{D}}(\hat{f}_{\widetilde{D}}) = \underbrace{(R_{\mathcal{D}}(\hat{f}_{\widetilde{D}}) - R_{\widetilde{\mathcal{D}}}(\hat{f}_{\widetilde{D}}))}_{\text{Term-1}} + \underbrace{R_{\widetilde{\mathcal{D}}}(\hat{f}_{\widetilde{D}})}_{\text{Term-2}}$, where **Term-1** transforms the evaluation of $\hat{f}_{\widetilde{D}}$ from clean distribution $\mathcal{D}$ to the pseudo noisy distribution $\widetilde{\mathcal{D}}$. **Term-2** is similar to the generalization error in Theorem 1 but the model is trained and evaluated on noisy distribution $\widetilde{\mathcal{D}}$. Both terms are analyzed as follows.

**Upper and Lower Bounds for Term-1**  Let $\eta(X) := \frac{1}{2} \sum_{i\in[K]} |\mathbb{P}(\widetilde{Y} = i|X) - \mathbb{P}(Y = i|X)|$, $e(X) := \mathbb{P}(Y \neq \widetilde{Y}|X)$ be the feature-dependent error rate, $\widetilde{A}_f(X) := \mathbb{P}(f(X) = \widetilde{Y}|X)$ be the accuracy of prediction $f(X)$ on noisy dataset $\widetilde{D}$. Denote their expectations (over $X$) by $\bar{\eta} := \mathbb{E}_X[\eta(X)]$, $\bar{e} := \mathbb{E}_X[e(X)]$, $\widetilde{A}_f = \mathbb{E}_X[\widetilde{A}_f(X)]$. To highlight that $\bar{\eta}$ and $\bar{e}$ depends on the noisy dataset $\widetilde{D}$ labeled by $\hat{f}_{D_L}$, we denote them as $\bar{\eta}(\hat{f}_{D_L})$ and $\bar{e}(\hat{f}_{D_L})$. Then we have:

**Lemma 1** (Bounds for Term-1). $(2\widetilde{A}_{\hat{f}_{\widetilde{D}}} - 1)\bar{e}(\hat{f}_{D_L}) \leq R_{\mathcal{D}}(\hat{f}_{\widetilde{D}}) - R_{\widetilde{\mathcal{D}}}(\hat{f}_{\widetilde{D}}) \leq \bar{\eta}(\hat{f}_{D_L})$.

Note the upper bound builds on $\bar{\eta}(\hat{f}_{D_L})$ while the lower bound relates to $\bar{e}(\hat{f}_{D_L})$. To compare two bounds and show the tightness, we consider the case where $Y|X$ is *confident*, i.e., each feature $X$ belongs to one particular true class $Y$ with probability 1, which is generally held in classification problems (Liu & Tao, 2015). Lemma 2 shows $\eta(X) = e(X)$ in this particular case.

**Lemma 2** ($\eta$ vs. $e$). *For any feature $X$, if $Y|X$ is confident, $\eta(X)$ is the error rate of the model prediction on $X$, i.e., $\exists i \in [K] : \mathbb{P}(Y = i|X) = 1 \Rightarrow \eta(X) = \mathbb{P}(\widetilde{Y} \neq Y|X) = e(X)$.*

**Upper bound for Term-2**  Denote by $\widetilde{Y}^*|X := \arg\max_{i\in[K]} \mathbb{P}(\widetilde{Y} = i|X)$ the Bayes optimal label on noisy distribution $\widetilde{\mathcal{D}}$. Following the proof for Theorem 1, we have:

**Lemma 3** (Bound for Term-2). *W. p. at least $1 - \delta$, $R_{\widetilde{\mathcal{D}}}(\hat{f}_{\widetilde{D}}) \leq \sqrt{\frac{2\log(2/\delta)}{N}} + \mathbb{P}(\widetilde{Y} \neq \widetilde{Y}^*)$.*

**Wrap-up**  Lemma 1 shows Term-1 is in the range of $[(2\widetilde{A}_{\hat{f}_{\widetilde{D}}} - 1)\bar{e}(\hat{f}_{D_L}), \bar{\eta}(\hat{f}_{D_L})]$. Lemma 2 informs us $\bar{\eta}(\hat{f}_{D_L}) = \bar{e}(\hat{f}_{D_L})$ in classification problems where $Y|X, \forall X$ are confident. With a *well-trained* model $\hat{f}_{\widetilde{D}}$ that learns the noisy distribution $\widetilde{\mathcal{D}}$, we have $\widetilde{A}_{\hat{f}_{\widetilde{D}}} = 1 - \epsilon$ and $\epsilon \to 0_+$, thus Term-1 is in the range of $[(1 - 2\epsilon)\bar{\eta}(\hat{f}_{D_L}), \bar{\eta}(\hat{f}_{D_L})]$, indicating *our bounds for Term-1 are tight*. Besides, noting Lemma 3 is derived following the same techniques as Theorem 1, we know both bounds have similar tightness. Therefore, by adding upper bounds for Term-1 and Term-2, we can upper bound the error of semi-supervised learning in Theorem 2, which has similar tightness to that in Theorem 1.

**Theorem 2** (Semi-supervised learning error). *Suppose the model trained with only $D_L$ has generalization error $\bar{\eta}'(\hat{f}_{D_L})$. With probability at least $1 - \delta$, the generalization error of semi-supervised learning on datasets $D_L \cup D_U$ is upper-bounded by*

$$R_{\mathcal{D}}(\hat{f}_{D_L \cup D_U}) \leq \underbrace{\bar{\eta}(\hat{f}_{D_L})}_{\text{Disparity due to baseline}} + \underbrace{\mathbb{P}(\widetilde{Y} \neq \widetilde{Y}^*)}_{\text{Sharpness of pseudo labels}} + \underbrace{\sqrt{\frac{2\log(2/\delta)}{N}}}_{\text{Data dependency}},$$

*where $\bar{\eta}(\hat{f}_{D_L}) := \bar{\eta}'(\hat{f}_{D_L}) \cdot N_U/N$ is the expected label error in the pseudo noisy dataset $\widetilde{D}$.*

**Takeaways**  Theorem 2 explains how disparate impacts in SSL are generated.

- Supervised error $\bar{\eta}'(\hat{f}_{D_L})$: the major source of disparity. The sub-population that generalizes well before SSL tends to have a lower SSL error. Namely, the rich get richer.
- Sharpness of noisy labels $\mathbb{P}(\widetilde{Y} \neq \widetilde{Y}^*)$: a minor source of disparity depends on how one processes pseudo-labels. This term is negligible if we sharpen the pseudo-label.
- Sample complexity $\sqrt{2\log(2/\delta)/N}$: disparity depends on the number of instances $N$ and their independence. Recall we assume ideal data augmentations to get $N$ in this term. There will be much less than $N$ *i.i.d.* instances with poor data augmentations. It would be a major source of disparity if data augmentations are poor.

## 4 BENEFIT RATIO: AN EVALUATION METRIC

Figure 1 demonstrates that SSL leads to disparate impacts of different sub-populations' accuracies, but it is still not clear how much that SSL benefits a certain sub-population. To quantify the disparate impacts of SSL, we propose a new metric called *benefit ratio*.

**Benefit Ratio** The benefit ratio $\mathsf{BR}(\mathcal{P})$ captures the normalized accuracy improvement of sub-population $\mathcal{P}$ after SSL, which depends on three classifiers, i.e., $\hat{f}_{D_L}$: (baseline) supervised learning only with a small labeled data $D_L$, $\hat{f}_D$: (ideal) supervised learning if the whole dataset $D$ has ground-truth labels, and $\hat{f}_{D_L \cup D_U}$: SSL with both labeled dataset $D_L$ and unlabeled dataset $D_U$. The test/validation accuracy of the above classifiers are $a_{\text{baseline}}(\mathcal{P})$, $a_{\text{ideal}}(\mathcal{P})$, and $a_{\text{semi}}(\mathcal{P})$, respectively. As a posterior evaluation of SSL algorithms, the benefit ratio $\mathsf{BR}(\mathcal{P})$ is defined as:

$$\mathsf{BR}(\mathcal{P}) = \frac{a_{\text{semi}}(\mathcal{P}) - a_{\text{baseline}}(\mathcal{P})}{a_{\text{ideal}}(\mathcal{P}) - a_{\text{baseline}}(\mathcal{P})}. \tag{3}$$

Let $\mathcal{P}^\diamond := \{\mathcal{P}_1, \mathcal{P}_2, \cdots\}$ be the set of all the concerned sub-populations. We formally define the Equalized Benefit Ratio as follows.

**Definition 1** (Equalized Benefit Ratio). *We call an algorithm achieving equalized benefit ratio if all the concerned sub-populations have the same benefit ratio:* $\mathsf{BR}(\mathcal{P}) = \mathsf{BR}(\mathcal{P}'), \forall \mathcal{P}, \mathcal{P}' \in \mathcal{P}^\diamond$.

Intuitively, a larger benefit ratio indicates more benefits from SSL. We have $\mathsf{BR}(\mathcal{P}) = 1$ when SSL performs as well as the corresponding fully-supervised learning. A negative benefit ratio indicates the poor population is hurt by SSL such that $a_{\text{semi}}(\mathcal{P}) < a_{\text{baseline}}(\mathcal{P})$, i.e., the poor get poorer as shown in Figure 1(b) (sub-population *dog*). It has the potential of providing guidance and intuitions for designing fair SSL algorithms on standard datasets with full ground-truth labels. Whether a fair SSL algorithm on one dataset is still fair on another dataset would be an interesting future work. In real scenarios without full supervisions, we may use some extra knowledge to estimate the highest achievable accuracy of each sub-population and set it as a proxy of the ideal accuracy $a_{\text{ideal}}(\mathcal{P})$.

**Theoretical Explanation** Recall we have error upper bounds for both supervised learning (Theorem 1) and semi-supervised learning (Theorem 2). Both bounds have similar tightness thus we can compare them and get a proxy of benefit ratio as $\widehat{\mathsf{BR}}(\mathcal{P}) := \frac{\sup{(R_{\mathcal{D}}(\hat{f}_{D_L \cup D_U | \mathcal{P}})) - \sup{(R_{\mathcal{D}}(\hat{f}_{D_L | \mathcal{P}}))}}}{\sup(R_{\mathcal{D}}(\hat{f}_{D | \mathcal{P}})) - \sup(R_{\mathcal{D}}(\hat{f}_{D_L | \mathcal{P}}))}$, where $\sup(\cdot)$ denotes the upper bound derived in Theorem 1 and Theorem 2, $\mathcal{P}$ is a sub-population, and $D | \mathcal{P}$ denotes the set of *i.i.d.* instances in $D$ that affect model generalization on $\mathcal{P}$. By assuming $\mathbb{P}(Y \neq Y^*) = \mathbb{P}(\widetilde{Y} \neq \widetilde{Y}^*)$ (both distributions have the same sharpness), we have:

**Corollary 1.** *The benefit ratio proxy for $\mathcal{P}$ is* $\widehat{\mathsf{BR}}(\mathcal{P}) = 1 - \frac{\bar{\eta}(\hat{f}_{D_L | \mathcal{P}})}{\Delta(N_\mathcal{P}, N_{\mathcal{P}_L})}$, *where* $\Delta(N_\mathcal{P}, N_{\mathcal{P}_L}) = \sqrt{\frac{2\log(2/\delta)}{N_{\mathcal{P}_L}}} - \sqrt{\frac{2\log(2/\delta)}{N_\mathcal{P}}}$, $N_\mathcal{P}$ *and* $N_{\mathcal{P}_L}$ *are the effective numbers of instances in $D | \mathcal{P}$ and $D_L | \mathcal{P}$.*

Corollary 1 shows the benefit ratio is negatively related to the error rate of baseline models and positively related to the number of *i.i.d.* instances after in SSL, which is consistent with our takeaways from Section 3. Note $N_\mathcal{P}$ and $N_{\mathcal{P}_L}$ may be larger than the corresponding sub-populations sizes if $\mathcal{P}$ shares information with other sub-population $\mathcal{P}'$ during training, e.g., a better classification of $\mathcal{P}'$ helps classify $\mathcal{P}$. It also informs us that SSL may have a negative effect on sub-population $\mathcal{P}$ if $\frac{\eta}{\Delta(N_\mathcal{P}, N_{\mathcal{P}_L})} > 1$, i.e., the benefit from getting more effective *i.i.d.* instances is less than the harm from wrong pseudo-labels. This negative effect indicates "the poor get poorer".

## 5 EXPERIMENTS

In this section, we first show the existence of disparate impact on several representative SSL methods and then discuss the possible treatment methods to mitigate this disparate impact.

**Settings** Three representative SSL methods, i.e., MixMatch (Berthelot et al., 2019b), UDA (Xie et al., 2019), and MixText (Chen et al., 2020), are tested on several image and text classification tasks. For image classification, we experiment on CIFAR-10 and CIFAR-100 datasets (Krizhevsky et al., 2009). We adopt the *coarse labels* (20 classes) in CIFAR-100 for training and test the performance for each *fine label* (100 classes). Thus our training on CIFAR-100 is a 20-class classification task and each coarse class contains 5 sub-populations. For text classification, we employ

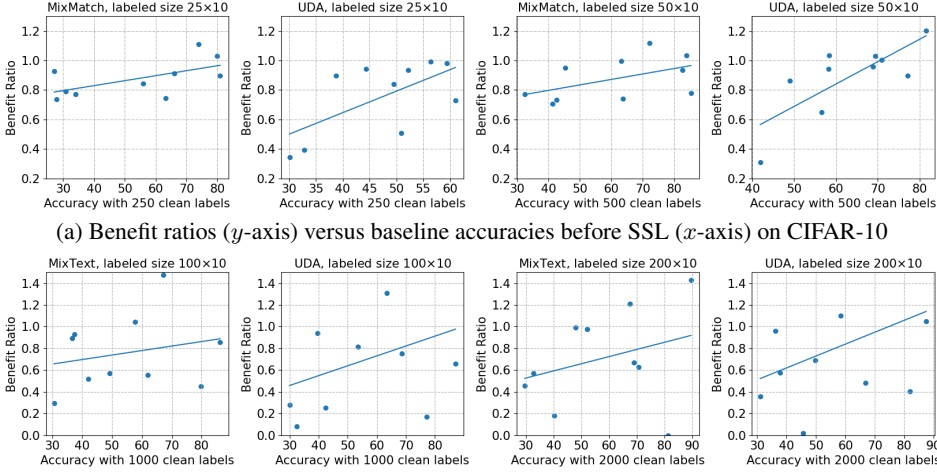

(a) Benefit ratios ($y$-axis) versus baseline accuracies before SSL ($x$-axis) on CIFAR-10

(b) Benefit ratios ($y$-axis) versus baseline accuracies before SSL ($x$-axis) on Yahoo! Answers

Figure 2: Benefit ratios across explicit sub-populations. Dot: Result of each label class. Line: Best linear fit of dots.

three datasets: Yahoo! Answers (Chang et al., 2008), AG News (Zhang et al., 2015) and Jigsaw Toxicity (Kaggle, 2018). Jigsaw Toxicity dataset contains both classification labels (one text comment is toxic or non-toxic) and a variety of sensitive attributes (e.g., race and gender information) of the identities that are mentioned in the text comment, which are fairness concerns in real-world applications. We defer more experimental details in Appendix C.

## 5.1 DISPARATE IMPACT EXISTS IN POPULAR SSL METHODS

We show that, even though the size of each sub-population is equal, disparate impacts exist in the model accuracy of different sub-populations, i.e., 1) *explicit sub-populations* such as classification labels in Figure 2 (and Figure 5 in Appendix C.2), and 2) *implicit sub-populations* such as fine-categories under coarse classification labels in Figure 3a, demographic groups including race in Figure 3b and gender (Figure 6 in Appendix C.3). All the experiments in this subsection adopt both a balanced labeled dataset and a balanced unlabeled dataset. Note we sample a balanced subset from the raw (unbalanced) Jigsaw dataset. Other datasets are originally balanced.

**Disparate impact across explicit sub-populations** In this part, we show the disparate impact on model accuracy across different classification labels on CIFAR-10, Yahoo! Answers, and AG News datasets. Detailed results on AG News can be found in Appendix C.2. In Figure 2a and 2b, we show the benefit ratios ($y$-axis) versus baseline accuracies before SSL ($x$-axis). From left to right, we show results with different sizes of labeled data: 25 per class to 50 on CIFAR-10 and 100 per class to 200 on Yahoo! Answers. Figure 2a and 2b utilized two SSL methods respectively (CIFAR-10: MixMatch & UDA; Yahoo! Answers: MixText & UDA).We statistically observe that the class labels with higher baseline accuracies have higher benefit ratios on both CIFAR-10 and Yahoo! Answers datasets. It means that "richer" classes benefit more from applying SSL methods than the "poorer" ones. We also observe that for some models with low baseline accuracy (left side of the $x$-axis), applying SSL results in rather low benefit ratios that are close to 0.

**Disparate impact across implicit sub-populations** We demonstrate the disparate impacts on model accuracy across different sub-populations on CIFAR-100 (fine-labels) and Jigsaw (race and gender) datasets. In Figure 3, we can statistically observe the disparate impact across different sub-populations on both datasets for three baseline SSL methods. Detailed results on Jigsaw with gender are shown in Appendix C.3. We again observe very similar disparate improvements as presented in Figure 2a and 2b - for some classes in CIFAR-100, this ratio can even go negative. Note the disparate impact on the demographic groups in Jigsaw raises fairness concerns in real-world applications.

## 5.2 MITIGATING DISPARATE IMPACT

Our analyses in Section 3 and Section 4 indicate the disparate impacts may be mitigated by balancing the supervised error $\bar{\eta}(\hat{f}_{D_L|\mathcal{P}})$ and the number of effective *i.i.d.* instances for different populations.

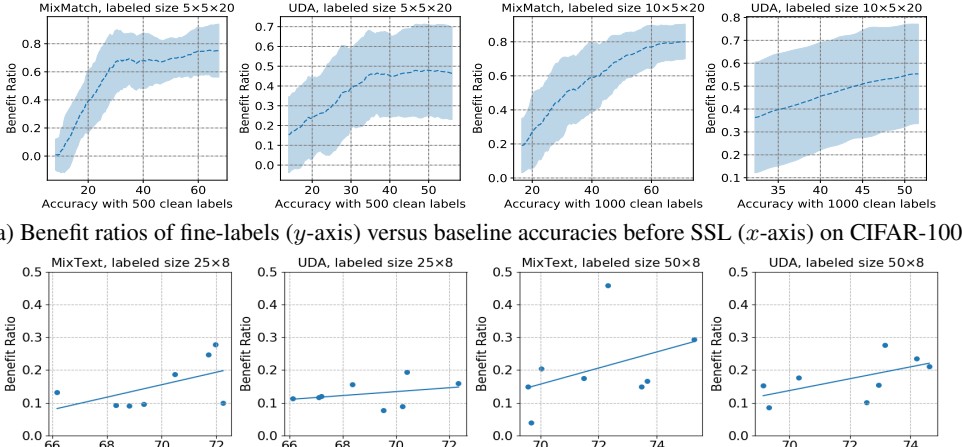

(a) Benefit ratios of fine-labels ($y$-axis) versus baseline accuracies before SSL ($x$-axis) on CIFAR-100

(b) Benefit ratios of different race attributes ($y$-axis) versus baseline accuracies before SSL on Jigsaw.

Figure 3: Benefit ratios across implicit sub-populations. Experiments on CIFAR-100 are ran 5 times for stability. Mean (dashed line) and standard deviation (shaded area) are plotted in (a), where points in the shaded area indicate the converged local optima with a non-negligible probability.

Table 1: Comparison on the benefit ratio (%) of all race identities & standard deviation (%) between different settings. $t\_asian$ ($nt\_asian$) is the benefit ratio of asian identity with "toxic" labels ("non-toxic" labels). *SD* denotes standard deviation. Negative benefit ratios are highlighted in red colors.

| [Jigsaw] Settings | $t\_asian$ | $t\_black$ | $t\_latin$ | $t\_white$ | $nt\_asian$ | $nt\_black$ | $nt\_latin$ | $nt\_white$ | SD |
|---|---|---|---|---|---|---|---|---|---|
| Unbalanced (400) | 24.71 | 21.76 | 28.21 | **-9.57** | 27.27 | **-8.33** | **-13.82** | 29.47 | 19.28 |
| Balance labeled (400) | 28.18 | 20.00 | 25.23 | 33.93 | 14.33 | **-11.29** | **-10.37** | 3.70 | 17.29 |
| Balance both (400) | 28.57 | 0.00 | 11.11 | 40.63 | 15.38 | 14.29 | 6.90 | **-7.41** | 15.29 |
| Balance both (800) | 13.04 | 25.00 | 7.69 | 24.00 | 3.85 | 10.71 | 16.67 | 20.00 | 7.64 |

We perform preliminary tests to check the effectiveness of the above findings in this subsection. We hope our analyses and experiments could inspire future contributions to disparity mitigation in SSL.

**Balancing and Collecting more labeled data** We firstly sample 400 *i.i.d.* instances from the raw (unbalanced) Jigsaw dataset and get the setting of Unbalanced (400). To balance the supervised error and effective instances for different sub-populations, an intuitive method is to balance the labeled data by reweighting, e.g., if the size of two sub-populations is 1:2, we simply sample instances from two sub-populations with a probability ratio of 2:1 to ensure all sub-population have the same weights in each epoch during training. Balance labeled (400) denotes only 400 labeled instances are reweighted. Balance both (400) means both labeled and unlabeled instances are balanced. Table 1 shows a detailed comparison on the benefit ratios between different race identities of the Jigsaw dataset and their standard deviations, where the first three rows denote the above three settings. We can observe that both the standard deviation and the number of negative benefit ratios become lower with more balanced settings, which demonstrates the effectiveness of the balancing treatment strategy, although there still exists a sub-population that has a negative benefit ratio, indicating an unfair learning result. To further improve fairness, as suggested in our analyses, we "collect" (rather add) another 400 *i.i.d.* labeled instances (800 in total) from the Jigsaw dataset, and show the result after balancing both labeled and unlabeled data in the last row of Table 3 (Balance both (800)). Both the standard deviation and the number of negative benefit ratios can be further reduced with more labeled data. More experimental results are on CIFAR-100 and Jigsaw (gender) datasets shown in Appendix C.4 (balancing) and Appendix C.5 (more data).

## 6 CONCLUSIONS

We have theoretically and empirically shown that the disparate impact (the "rich" sub-populations get richer and the "poor" ones get poorer) exists universally for a broad family of SSL algorithms. We have also proposed and studied a new metric *benefit ratio* to facilitate the evaluation of SSL. We hope our work could inspire more efforts towards mitigating the disparate impact and encourage a multifaceted evaluation of existing and future SSL algorithms.

ACKNOWLEDGMENTS

This work is partially supported by the National Science Foundation (NSF) under grant IIS-2007951, IIS-2143895, and the NSF FAI program in collaboration with Amazon under grant IIS-2040800.

ETHICS STATEMENT

In recent years, machine learning methods are widely applied in the real world across a broad variety of fields such as face recognition, recommendation system, information retrieval, medical diagnosis, and marketing decision. The trend is likely to continue growing. However, it is noted that these machine learning methods are susceptible to introducing unfair treatments and can lead to systematic discrimination against some demographic groups. Our paper echoes this effort and looks into the disparate model improvement resulted from deploying a semi-supervised learning (SSL) algorithm.

Our work uncovers the unfairness issues in semi-supervised learning methods. More specifically, we theoretically and empirically show the disparate impact exists in the recent SSL methods on several public image and text classification datasets. Our results provide understandings of the potential disparate impacts of an SSL algorithm and help raise awareness from a fairness perspective when deploying an SSL model in real-world applications. Our newly proposed metric benefit ratio contributes to the literature a new measure to evaluate the fairness of SSL. Furthermore, we discuss two possible treatment methods: balancing and collecting more labeled data to mitigate the disparate impact in SSL methods. We have carefully studied and presented their effectiveness in mitigating unfairness in the paper. Our experimental results, such as Table 1, demonstrate the disparate impacts across demographic groups (e.g., gender, race) naturally exist in SSL. We are not aware of the misuse of our proposals, but we are open to discussions.

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

# Appendix

The Appendix is organized as follows.

## A  THEORETICAL RESULTS

### A.1  TERM-1 UPPER BOUND

$$
\begin{aligned}
& R_{\mathcal{D}}(f) - R_{\widetilde{\mathcal{D}}}(f) \\
&= \int_X \left( \mathbb{P}(f(X) \neq Y | X) - \mathbb{P}(f(X) \neq \widetilde{Y} | X) \right) \mathbb{P}(X)\, dX \\
&= \int_X \left( \mathbb{P}(f(X) = \widetilde{Y} | X) - \mathbb{P}(f(X) = Y | X) \right) \mathbb{P}(X)\, dX \\
&\leq \frac{1}{2} \int_X \left( \left| \mathbb{P}(f(X) = \widetilde{Y} | X) - \mathbb{P}(f(X) = Y | X) \right| + \left| \mathbb{P}(f(X) \neq \widetilde{Y} | X) - \mathbb{P}(f(X) \neq Y | X) \right| \right) \mathbb{P}(X)\, dX \\
&\overset{(a)}{=} \int_X \mathrm{TD}(\widetilde{Y}_f(X) || Y_f(X)) \mathbb{P}(X)\, dX \\
&\overset{(b)}{\leq} \int_X \mathrm{TD}(\widetilde{Y}(X) || Y(X)) \mathbb{P}(X)\, dX \\
&= \frac{1}{2} \int_X \sum_{i \in [K]} \left| \mathbb{P}(\widetilde{Y} = i | X) - \mathbb{P}(Y = i | X) \right| \mathbb{P}(X)\, dX \\
&= \int_X \eta(X) \mathbb{P}(X)\, dX \\
&= \eta
\end{aligned}
$$

where in **equality (a)**, given model $f$ and feature $X$, we can treat $\widetilde{Y}_f(X)$ as a Bernoulli random variable such that

$$
\mathbb{P}(\widetilde{Y}_f(X) = +) = \mathbb{P}(f(X) = \widetilde{Y} | X) \text{ and } \mathbb{P}(\widetilde{Y}_f(X) = -) = \mathbb{P}(f(X) \neq \widetilde{Y} | X).
$$

Then according to the definition of total variation of two distributions, i.e.,

$$
\mathrm{TD}(P || Q) := \frac{1}{2} \int_u |\frac{dP}{du} - \frac{dQ}{du}| du,
$$

we can summarize the integrand as the total variation between $\widetilde{Y}_f(X)$ and $Y_f(X)$.

**Inequality (b)** holds due to the data processing inequality since the probabilities $[\mathbb{P}(\widetilde{Y} = f(X)), \mathbb{P}(\widetilde{Y} \neq f(X))]$ are generated by $[\mathbb{P}(\widetilde{Y} = i), \forall i \in [K]]$, and the probabilities $[\mathbb{P}(Y = f(X)), \mathbb{P}(Y \neq f(X))]$ are generated by $[\mathbb{P}(Y = i), \forall i \in [K]]$.

$\eta(X)$ is the accuracy of labels on feature $X$ (in the cases specified in Lemma 2).

### A.2  TERM-1 LOWER BOUND

Let $e(X) := \mathbb{P}(Y \neq \widetilde{Y} | X)$ be the feature-dependent error rate, $\widetilde{A}_f(X) := \mathbb{P}(f(X) = \widetilde{Y} | X)$ be the accuracy of prediction $f(X)$ on noisy dataset $\widetilde{D}$. Note $e(X)$ is independent of $\widetilde{A}_f(X)$. Denote their expectations (over $X$) by $\bar{e} := \mathbb{E}_X[e(X)]$, $\widetilde{A}_f = \mathbb{E}_X[\widetilde{A}_f(X)]$. We have:

$$R_{\mathcal{D}}(f) - R_{\widetilde{\mathcal{D}}}(f)$$

$$= \int_X \left( \mathbb{P}(f(X) \neq Y|X) - \mathbb{P}(f(X) \neq \widetilde{Y}|X) \right) \mathbb{P}(X)$$

$$= \int_X \left( \mathbb{P}(f(X) = \widetilde{Y}|X) - \mathbb{P}(f(X) = Y|X) \right) \mathbb{P}(X)$$

$$= \int_X \left( \underbrace{\mathbb{P}(f(X) = \widetilde{Y}|f(X) = \widetilde{Y}, X)}_{\text{w.p. 1}} - \underbrace{\mathbb{P}(f(X) = Y|f(X) = \widetilde{Y}, X)}_{\mathbb{P}(\widetilde{Y}=Y|X)} \right) \mathbb{P}(f(X) = \widetilde{Y}|X)\mathbb{P}(X)$$

$$+ \int_X \left( \underbrace{\mathbb{P}(f(X) = \widetilde{Y}|f(X) \neq \widetilde{Y}, X)}_{\text{w.p. 0}} - \mathbb{P}(f(X) = Y|f(X) \neq \widetilde{Y}, X) \right) \mathbb{P}(f(X) \neq \widetilde{Y}|X)\mathbb{P}(X)$$

$$= \int_X \underbrace{\left( 1 - \mathbb{P}(\widetilde{Y} = Y|X) \right)}_{\text{denoted by } e(X)} \underbrace{\mathbb{P}(f(X) = \widetilde{Y}|X)}_{\text{denoted by } \widetilde{A}_f(X)} \mathbb{P}(X)$$

$$+ \int_X \left( 0 - \mathbb{P}(f(X) = Y|f(X) \neq \widetilde{Y}, X) \right) \mathbb{P}(f(X) \neq \widetilde{Y}|X)\mathbb{P}(X)$$

$$= \int_X e(X)\widetilde{A}_f(X)\mathbb{P}(X)$$

$$- \int_X \left( \mathbb{P}(f(X) = Y|f(X) \neq \widetilde{Y}, Y = \widetilde{Y}, X)\mathbb{P}(Y = \widetilde{Y}|f(X) \neq \widetilde{Y}, X) \right) \mathbb{P}(f(X) \neq \widetilde{Y}|X)\mathbb{P}(X)$$

$$- \int_X \left( \underbrace{\mathbb{P}(f(X) = Y|f(X) \neq \widetilde{Y}, Y \neq \widetilde{Y}, X)}_{\leq 1} \mathbb{P}(Y \neq \widetilde{Y}|f(X) \neq \widetilde{Y}, X) \right) \mathbb{P}(f(X) \neq \widetilde{Y}|X)\mathbb{P}(X)$$

$$\geq \int_X e(X)\widetilde{A}_f(X)\mathbb{P}(X)$$

$$- \int_X \left( 0 \cdot \mathbb{P}(Y = \widetilde{Y}|f(X) \neq \widetilde{Y}, X) \right) \mathbb{P}(f(X) \neq \widetilde{Y}|X)\mathbb{P}(X)$$

$$- \int_X \left( 1 \cdot \underbrace{\mathbb{P}(Y \neq \widetilde{Y}|f(X) \neq \widetilde{Y}, X)}_{=\mathbb{P}(Y \neq \widetilde{Y}|X) \text{ due to independency}} \right) \mathbb{P}(f(X) \neq \widetilde{Y}|X)\mathbb{P}(X)$$

$$= \int_X e(X)\widetilde{A}_f(X)\mathbb{P}(X) - \int_X \underbrace{\mathbb{P}(Y \neq \widetilde{Y}|X)}_{\text{denoted by } e(X)} \underbrace{\mathbb{P}(f(X) \neq \widetilde{Y}|X)}_{\text{denoted by } 1 - \widetilde{A}_f(X)} \mathbb{P}(X)$$

$$= \int_X e(X)(2\widetilde{A}_f(X) - 1)\mathbb{P}(X)$$

$$= (2\widetilde{A}_f - 1)e.$$

## A.3  TERM-2 UPPER BOUND

We adopt the same technique to prove Theorem 1 and Lemma 3. The proof follows Liu & Guo (2020). We prove Theorem 1 as follows.

Denote the expected error rate of classifier $f$ on distribution $\mathcal{D}$ by

$$R_{\mathcal{D}}(f) := \mathbb{E}_{\mathcal{D}}[\mathbb{1}(f(X), Y)].$$

Let $\hat{f}_D$ denote the classifier trained by minimizing 0-1 loss with clean dataset $D$, i.e.,

$$\hat{f}_D := \arg\min_f \hat{R}_D(f),$$

where

$$\hat{R}_D(f) := \frac{1}{N} \sum_{n\in[N]} \mathbb{1}(f(x_n), y_n).$$

The Bayes optimal classifier is denoted by

$$f_{\mathcal{D}}^* := \arg\min_f R_{\mathcal{D}}(f).$$

The expected error given by $f_{\mathcal{D}}^*$ is written as

$$R^* := R_{\mathcal{D}}(f_{\mathcal{D}}^*).$$

Denote by $Y^*|X := \arg\max_{i\in[K]} \mathbb{P}(Y|X)$ the Bayes optimal label on clean distribution $\mathcal{D}$. With probability at least $1 - \delta$, we have:

$$
\begin{aligned}
&R_{\mathcal{D}}(\hat{f}_D) - R_{\mathcal{D}}(f_{\mathcal{D}}^*) \\
=& \hat{R}_D(\hat{f}_D) - \hat{R}_D(f_{\mathcal{D}}^*) + \left( R_{\mathcal{D}}(\hat{f}_D) - \hat{R}_D(\hat{f}_D) \right) + \left( \hat{R}_D(f_{\mathcal{D}}^*) - R_{\mathcal{D}}(f_{\mathcal{D}}^*) \right) \\
\overset{(a)}{\leq}& 0 + 2\max_{f\in\mathcal{F}} |\hat{R}_D(f) - R_{\mathcal{D}}(f)| \\
\overset{(b)}{\leq}& \sqrt{\frac{2\log(2/\delta)}{N}},
\end{aligned}
$$

where inequality (a) holds since 1) $\hat{R}_D(\hat{f}_D)$ achieves the minimum empirical risk according to its definition, thus $\hat{R}_D(\hat{f}_D) - \hat{R}_D(f_{\mathcal{D}}^*) \leq 0$; 2) each of the following two terms will be no greater than the corresponding maximal gap $\max_{f\in\mathcal{F}} |\hat{R}_D(f) - R_{\mathcal{D}}(f)|$. Inequality (b) holds due to the Hoeffding's inequality, i.e., with probability at least $1 - \delta$,

$$\max_{f\in\mathcal{F}} |\hat{R}_D(f) - R_{\mathcal{D}}(f)| \leq \sqrt{\frac{\log(2/\delta)}{2N}}.$$

Noting $R_{\mathcal{D}}(f_{\mathcal{D}}^*) = \mathbb{P}(Y \neq Y^*)$, we can prove Theorem 1. But substituting $\mathcal{D}$ for $\widetilde{\mathcal{D}}$, we can also prove Lemma 3.

### A.4 PROOF FOR LEMMA 2

*Proof.* Note

$$\mathbb{P}(\widetilde{Y} \neq Y|X) = 1 - \mathbb{P}(\widetilde{Y} = Y|X) = 1 - \sum_{i\in[K]} \mathbb{P}(\widetilde{Y} = i|X)\mathbb{P}(Y = i|X).$$

and

$$\eta(X) := \frac{1}{2} \sum_{i\in[K]} |\mathbb{P}(\widetilde{Y} = i|X) - \mathbb{P}(Y = i|X)|.$$

Assume $\mathbb{P}(Y = i'|X) = 1$. We have $\mathbb{P}(\widetilde{Y} \neq Y|X) = 1 - \mathbb{P}(\widetilde{Y} = i'|X)$ and

$$\eta(X) := \frac{1}{2} \left( 1 - \mathbb{P}(\widetilde{Y} = i'|X) + \sum_{i\in[K], i\neq i'} \mathbb{P}(\widetilde{Y} = i|X) \right) = 1 - \mathbb{P}(\widetilde{Y} = i'|X).$$

$\square$

### A.5 PROOF FOR COROLLARY 1

$$
\begin{aligned}
\widehat{\mathsf{BR}}(\mathcal{P}) &= \frac{\eta + \sqrt{\frac{2\log(2/\delta)}{N_{\mathcal{P}}}} + \mathbb{P}(\widetilde{Y} \neq \widetilde{Y}^*) - \sqrt{\frac{2\log(2/\delta)}{N_{\mathcal{P}_L}}} - \mathbb{P}(Y \neq Y^*)}{\sqrt{\frac{2\log(2/\delta)}{N_{\mathcal{P}}}} - \sqrt{\frac{2\log(2/\delta)}{N_{\mathcal{P}_L}}}} \\
&= \frac{\Delta(N_{\mathcal{P}}, N_{\mathcal{P}_L}) - \eta}{\Delta(N_{\mathcal{P}}, N_{\mathcal{P}_L})}.
\end{aligned}
$$

## B  MATTHEW EFFECT

We further illustrate the Matthew effect when the initial labeled dataset is small and unbalanced. In Figure 4, we show the change of test accuracy due to SSL. We test two SSL algorithms on CIFAR-10: MixMatch (Berthelot et al., 2019b) (left panel) and UDA (Xie et al., 2019) (right panel), where each label class is treated as a sub-population. There are 20 labeled instances in each of the first 5 classes ($20 \times 5$), 10 labeled instances in each of the remaining classes ($10 \times 5$). The remaining instances in the first 5 classes and half of the remaining instances in the other 5 classes are used as unlabeled data, which ensures the unlabeled dataset has the same ratios among sub-populations as the labeled one. Figure 4 shows two "poor" sub-populations get poorer by using MixMatch and "four" sub-populations get poorer by using UDA, which consolidate the observation of Matthew effect of SSL. Note we also find the observation "the poor getting poorer" does not always happen when there are more labeled instances and a more balanced dataset. See more discussions in the next subsection.

### B.1  MORE EXAMPLES

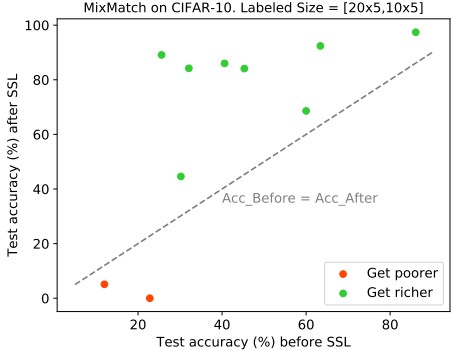 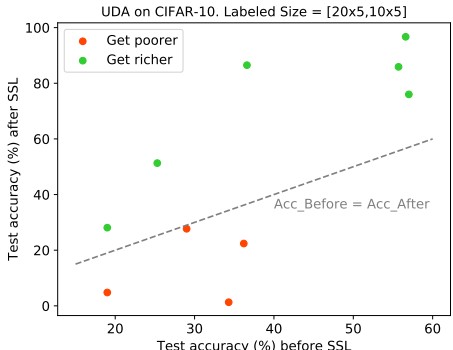

Figure 4: Disparate impacts in the model accuracy of SSL. MixMatch (Berthelot et al., 2019b) (left panel) and UDA (Xie et al., 2019) (right panel) are applied on CIFAR-10 with $150$ clean labels. Dashed line: The threshold indicating the test accuracy does not change before and after SSL. Red dots: Sub-populations that get poorer after SSL. Green dots: Sub-populations that get richer after SSL.

### B.2  DISCUSSIONS

We do observe some "poor" classes may benefit from SSL, e.g., class "dog" in Figure 1(a), and the dots that have initial test accuracies smaller than $0.3$. We are aware of that, for most cases, we do only observe "the rich getting richer". Intuitively, the phenomenon that several poor classes also benefit from SSL is mainly due to two reasons:

- In a resource-constrained scenario, "the rich getting richer" is likely to happen with "the poor getting poorer". However, benefiting from the large capacity of deep neural networks and powerful SSL algorithms, the available "resource" after SSL is usually greater than before SSL. Thus the poor may not always get poorer.
- Current popular SSL algorithms, such as MixMatch (Berthelot et al., 2019b) and UDA (Xie et al., 2019), are very powerful. It is reasonable to believe the re-labeling procedures in each epoch in these algorithms could reduce the disparate impact compared to the two-iteration scenario considered in our theoretical analysis. This may also explain why "some poor classes are adequately benefited." Although there are some exceptions, our Figure 2 and Figure 3 can still show that there are non-negligible disparate impacts including both "the rich get richer" and "the poor get poorer" in current state-of-the-art (SOTA) SSL algorithms. Besides, Figure 3(a) and Table 1 also show that some classes have negative benefit ratios, indicating "the poor get poorer".

Table 2: Dataset statistics and train/test splitting. The number of train and test samples means the number of data per class.

| Datasets | Label Type | Number of classes | Number of train samples | Number of test samples |
|---|---|---|---|---|
| CIFAR-10 | Image | 10 | 5000 | 1000 |
| CIFAR-100 | Image | 100 | 500 | 100 |
| Yahoo! Answer | Question Answering | 10 | 140000 | 5000 |
| AG News | News | 4 | 30000 | 1900 |

## C  MORE EXPERIMENTS

### C.1  EXPERIMENT SETTINGS

**Explicit sub-populations: classification labels**   In this section, we show the statistics of datasets where we conducted the experiments to demonstrate the disparate impact on the explicit sub-populations (classification labels) in Table 2. In addition, the train/valid/test splitting in the image datasets (CIFAR-10 and CIFAR-100) is 45000:5000:10000. As for the splitting in the text datsets (Yahoo! Answers and AG News), we follow the setting in (Chen et al., 2020).

**Implicit sub-populations: fine-categories under coarse classification labels**   In this section, we provide a description about coarse labels in CIFAR-100 dataset when we conducted the experiments to demonstrate the disparate impact on the implicit sub-populations: fine-categories under coarse classification labels. In CIFAR-100, its 100 classes can be categorized into 20 superclasses. Each image can have a "fine" label (the original class) and a "coarse" label (the superclass). The list of superclasses in the CIFAR-100 is as follows: aquatic mammals, fish , flowers orchids, food containers bottles, fruit and vegetables, household electrical devices, household furniture bed, large carnivores, large man-made outdoor things, large natural outdoor scenes, large omnivores and herbivores, medium-sized mammals, non-insect invertebrates, people, reptiles, small mammals, trees, vehicles 1, and vehicles 2.

**Implicit sub-populations: demographic groups (race & gender)**   In this section, we describe the detailed data distribution before and after balancing methods about implicit sub-populations: demographic groups (race & gender on Jigsaw dataset) . The identities in the race sub-population includes 'asian', 'black', 'latin', and 'white'. In each identity, we consider different classification labels (toxic and non-toxic). Therefore, we have eight identities in the race sub-population. The numbers of samples on these eight race identities in the unbalanced Jigsaw dataset are 'asian (toxic): 466; black (toxic): 1673; latin (toxic): 282; white (toxic): 3254; asian (non-toxic): 633; black (non-toxic): 1879; latin (non-toxic): 511; white (non-toxic): 2652'. The numbers of samples on these eight race identities in the balanced Jigsaw dataset are 'asian (toxic): 1419; black (toxic): 1419; latin (toxic): 1419; white (toxic): 1419; asian (non-toxic): 1419; black (non-toxic): 1419; latin (non-toxic): 1419; white (non-toxic): 1419'. Similar to race, we also have eight identities in the gender sub-population by considering four different gender types which are 'male', 'female', 'male femal' and 'transgender' and two classification labels (toxic and non-toxic). The numbers of samples on these eight gender identities in the unbalanced Jigsaw dataset are 'male (toxic): 5532; female (toxic): 4852; male female (toxic): 1739; transgender (toxic): 405; male (non-toxic): 5167; female (non-toxic): 4985; male female (non-toxic): 2002; transgender (non-toxic): 374'. The numbers of samples on these eight gender identities in the balanced Jigsaw dataset are 'male (toxic): 3142; female (toxic): 3142; male female (toxic): 3142; transgender (toxic): 3142; male (non-toxic): 3142; female (non-toxic): 3142; male female (non-toxic): 3142; transgender (non-toxic): 3142'. In either race or gender, Jigsaw dataset contains more than four identities, we set the number of identities to 4 due to the relatively small number of other identities. In addition, the ratio of 'train:valid:test' on either race or gender sub-population case is '8:1:1'.

## C.2   MORE EXPERIMENTAL RESULTS FOR DISPARATE IMPACT ACROSS DIFFERENT CLASSES ON AG NEWS

In this work, we also conducted experiments on the AG News text classification dataset. Figure 5 demonstrates the disparate impact on model accuracy across different classification labels on the AG News dataset.

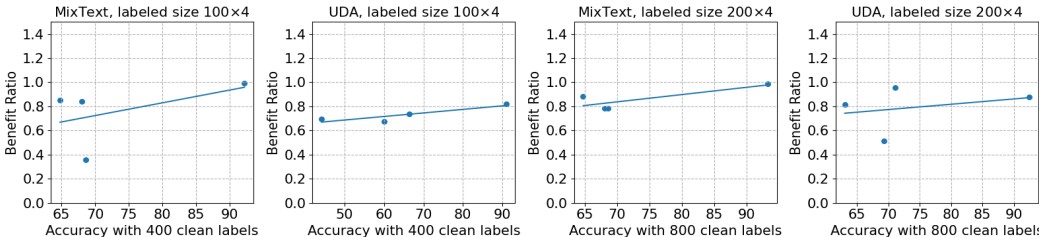

Figure 5: Benefit ratios of different class labels ($y$-axis) versus baseline accuracies before SSL ($x$-axis) on AG News

## C.3   MORE EXPERIMENTAL RESULTS FOR DISPARATE IMPACT ACROSS DIFFERENT SUB-POPULATIONS ON JIGSAW (GENDER)

In this work, we also conducted experiments on the Jigsaw Toxicity text classification dataset with gender sub-populations. Figure 6 shows the disparate impact on model accuracy across different sub-populations (gender) on the Jigsaw Toxicity dataset.

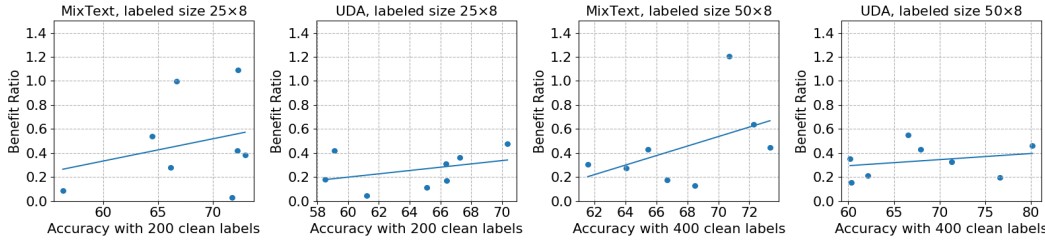

Figure 6: Benefit ratios of different gender attributes ($y$-axis) versus baseline accuracies before SSL ($x$-axis) on Jigsaw.

## C.4   MORE EXPERIMENTAL RESULTS OF "BALANCING" TREATMENT TO DISPARATE IMPACT

In this section, we show more experimental results of "Balancing" treatment to disparate impact described in Section 5.2. In Table 3, we compare the changes of mean and standard deviation on the model accuracy and our proposed benefit ratio in three settings: (a) SSL with unbalanced data → (b) SSL with balanced labeled datasets only (by reweighting) → (c) SSL with both balanced labeled & unlabeled datasets. Setting (c) is not practical due to the lack of labels in unlabeled data but it shows the best performance that this simple reweighting can achieve. Note that *CIFAR-100 1:2 Accuracy* denotes the accuracy of the case *CIFAR-100 1:2*, where CIFAR-100 1:2 means 40 instances per class for the first 50 fine classes, 20 instances per class for the next 50 fine classes. Besides, *Jigsaw (1.4:4.5:1:7.5) means that the ratios of different sub-populations' sizes are 1.4:4.5:1:7.5*. Table 3 shows reweighting unbalanced dataset helps improve the overall performance (mean) and reduce the disparate impacts (standard deviation).

## C.5   MORE EXPERIMENTAL RESULTS OF "COLLECTING MORE LABELED DATA" TREATMENT TO DISPARATE IMPACT

In this subsection, we show more experimental results of "Collecting more labeled data" treatment to disparate impact described on Section 5.2. Table 4 demonstrates the changes of mean and standard

Table 3: Balance samples with reweighting. We sample unbalanced CIFAR-100 (1:2 means 40 instances per class for the first 50 fine classes, 20 instances per class for the next 50 fine classes). Jigsaw is naturally unbalanced. For race, we consider asian, black, latin, and white identities. For gender, we consider male, female, male female, and transgender identities. The rounded ratios on the number of samples between different identities in race and gender are listed in the table.

| Datasets | *Mean* | *Standard Deviation* |
|---|---|---|
| CIFAR-100 1:2 Accuracy | 69.91 → 69.98 → **69.99** | 22.26 → 21.48 → **21.46** |
| CIFAR-100 1:5 Accuracy | 61.90 → 62.76 → 63.21 | 29.50 → 27.73 → 26.97 |
| CIFAR-100 1:2 Benefit Ratio | 55.16 → 55.19 → **56.09** | 20.85 → 20.16 → **19.17** |
| CIFAR-100 1:5 Benefit Ratio | 44.67 → 49.47 → 47.40 | 37.07 → 33.82 → 32.18 |
| Jigsaw (race) (1.4:4.5:1:7.5) Accuracy | 66.71 → 67.25 → **67.88** | 25.64 → 21.33 → 17.84 |
| Jigsaw (gender) (13.7:12.6:4.8:1) Accuracy | 66.03 → 66.74 → 67.17 | 12.90 → 9.75 → **8.68** |
| Jigsaw (race) (1.4:4.5:1:7.5) Benefit Ratio | 12.46 → 13.09 → **13.67** | 19.28 → 17.29 → **15.29** |
| Jigsaw (gender) (13.7:12.6:4.8:1) Benefit Ratio | 11.90 → 12.18 → 12.54 | 88.52 → 50.67 → 34.24 |

deviation on model accuracy and benefit ratio with the varying number (grow exponentially from left to right) of labeled data on CIFAR-100 and Jigsaw (race & gender). As shown in Table 4, the mean accuracy tends to be higher and the standard deviation becomes lower when more labeled data is utilized to train the model, which shows the effectiveness of collecting more labeled data.

Table 4: Collect more data. $a \to b \to c \to d$: stands for the change of mean or standard deviation with different sizes of labeled dataset. For CIFAR-100, the sizes are $5 \times 100, 10 \times 100, 20 \times 100, 40 \times 100$. For Jigsaw (both race and gender), the sizes are $25 \times 8, 50 \times 8, 100 \times 8, 150 \times 8$.

| Datasets | *Mean* | *Standard Deviation* |
|---|---|---|
| CIFAR-100 Accuracy | 52.87 → 60.93 → 68.01 → **74.69** | 32.38 → 28.59 → 23.52 → **16.76** |
| CIFAR-100 Benefit Ratio | 48.86 → 54.01 → 56.40 → **66.20** | 33.39 → 28.62 → 22.11 → **16.87** |
| Jigsaw (race) Accuracy | 64.88 → 67.88 → 72.25 → 73.50 | 29.55 → 17.84 → 9.93 → 6.86 |
| Jigsaw (gender) Accuracy | 64.50 → 67.17 → 73.50 → **74.83** | 20.15 → 8.68 → 6.41 → **5.99** |
| Jigsaw (race) Benefit Ratio | 12.09 → 13.67 → 14.52 → 15.12 | 18.29 → 15.29 → 11.69 → **7.64** |
| Jigsaw (gender) Benefit Ratio | 7.80 → 12.54 → 22.46 → **25.81** | 41.29 → 34.24 → 27.30 → 13.08 |

