# OpenReview forum: "The Rich Get Richer: Disparate Impact of Semi-Supervised Learning"
_ICLR.cc/2022/Conference — ICLR 2022 Poster_

### Official Review · Reviewer_x3NJ · 2021-10-29

**Correctness:** 3
**Technical Novelty And Significance:** 2
**Empirical Novelty And Significance:** 3
**Recommendation:** 6
**Confidence:** 4

**Main Review:**

## Strength
- Disparate impact on the constituent sub-population when training through SSL is an important problem.
- Their theoretical analysis is sound.

## Weakness

**Matthew effect: Rich getting richer and poor getting poorer**
I do not see the poor getting poorer part of their claim.
If poor gets poorer we should see the test accuracy of *dog* only decreasing with SSL in Fig 1 (b).
Besides the 'dog' class, no other "poor" sub-population got worser with training in the same figure, for e.g. see *horse, ship, truck* classes.
Similarly, in Fig 2, 3, we should have seen negative BR for the poor population if poor gets poorer, but that is not the case in any of the sixteen plots; all the BR values are positve.
Although, I agree that SSL does not benefit all the sub-population equally, and that "rich" sub-population gain owing to lower label noise in the induced labels on the unsupervised data.

**What is the fairness concern here?**
SSL training benefit some/rich sub-population more than the others.
As a result, it improves the average accuracy without hurting the worst group accuracy (since BR>=0 for all groups).
I would not call out SSL as unfair unless the poor population gets even worser, i.e. the rich groups get richer at the expense of poor groups.
The authors should be more formal when discussing the fairness aspect and explain depreciating population disparity in the context of a fairness metric.

**Analysis**
Is not $\bar\eta'(\hat f_{D_L})$ simply the risk with clean data, i.e. $R_{D}(\hat f_{D_L})$? This is so because both measure the expected disagreement of the prediction to the label (in the confident case of Lemma 2).
Replacing $\eta$ with $R_D$ in Thm 2, we have:

$$
R_D(\hat f_{D_L\cup D_U}) \leq \frac{N_U}{N}R_D(\hat f_{D_L}) + \text{<other terms>}
$$

When $N_U\rightarrow \infty, i.e. N_U\approx N$, we have $R_D(\hat f_{D_L\cup D_U}) \leq R_D(\hat f_{D_L}) + \Pr(\tilde Y\neq \tilde Y^*)$, and as a result the upper bound for SSL case is worse than the usual case.
The generalization error bound of Thm 2 therefore cannot explain the improved performance on a subset of population.
Please clarify if I am missing something.

Since the generalization upper bound of Thm. 2 is used for qualitative judgment on the effect of SSL, its sanity should be established I believe.

**More technical details needed**
I found Section 5.2 hard to understand even after consulting the Appendix.
Description for Table 1 numbers is unclear.
What does each of the settings mean? What is balanced: group population or label population? Where is the balancing, in labeled or unlabeled?

Somewhere in the intro, the authors mention BR could even be greater than 1, and we also see that in some of the BR plots (Fig 2 a,b), for the richer classes. I do not understand how.
If the ideal population size is at least N_L+N_U, any BR value greater than one is only due to the empirical error in accuracy estimation. Please clarify.


## Minor
- In the last line of A.3, Hoeffding's ineq. does not give uniform deviation bound unlike what was stated there.
- The upper bound of Thm 1 contains constants that go with the $1/\sqrt{n}$ term, such as the maximum possible value of loss. But the order remains the same.
- In Tab.3 of appendix B.4, need to explain what 1:2 Benefit ratio, 1:5 accuracy etc. mean.


**Summary Of The Paper:**

The paper claims semi-supervised learning (SSL) does not uniformly improve performance on all the constituent (latent) sub-population of the group.
More alarmingly, they claim that the SSL improves the performance most on the groups that already fare well (before SSL) with deteriorates on groups that are already worse.
They argue intuitively and theoretically for the likely cause.
With text, image datasets and three SSL algorithms, they provide empirical evidence that can support their main claim.
Finally, they briefly discuss strategies that could avoid such disparate impact of SSL.

**Summary Of The Review:**

The prime objective of the paper is to demonstrate the Matthew effect of SSL and its fairness consequences.
As described in more detail above, I have concerns regarding both the aspects of their contribution.

---

> ### Author Response · Authors · 2021-11-18
> **Response to Reviewer x3NJ (part 1)**
>
> Thanks for reviewing our paper and providing valuable comments. We would like to clarify that, in some cases, we do only observe "the rich get richer." Consequently, we did not mention the other half, "the poor get poorer," in our title. On the other hand, we observe severe fairness issues in datasets such as Jigsaw when the sub-population relates to race (Table 1). Your concerns are addressed in detail as follows.
>
> **Question 1: Matthew effect: Rich getting richer and poor getting poorer.**
> I do not see the poor getting poorer part of their claim. If poor gets poorer we should see the test accuracy of dog only decreasing with SSL in Fig 1 (b). Besides the 'dog' class, no other "poor" sub-population got worse with training in the same figure, for e.g. see horse, ship, truck classes. Similarly, in Fig 2, 3, we should have seen negative BR for the poor population if poor gets poorer, but that is not the case in any of the sixteen plots; all the BR values are positive. Although, I agree that SSL does not benefit all the sub-population equally, and that "rich" sub-population gain owing to lower label noise in the induced labels on the unsupervised data.
>
> **Response 1:**
>
> We'd like to point out that our title only suggests one direction of the Matthew effect: the rich get richer. We are aware that the observations for "poor get poorer" is less than "richer get richer": for most of our observations. Nonetheless, as mentioned in Motivating Example (shown in Introduction), the full Matthew effect is observed in the extreme or unbalanced cases, such as in Figure 1(b). **The test accuracy of class dog does decrease during SSL in Figure 1(b).** Figure 1(b) shows that class dog has **an initial test accuracy of about $5\%$** at epoch 0, but it **drops to almost $0\%$ accuracy** at epoch 250. This observation supports the claim "the poor get poorer". Intuitively, an unbalanced dataset is more likely to have a disparate impact. That's why we observe "the poor get poorer" in Figure 1(b) but not in Figure 1(a).
>
> Our Figure 2 and Figure 3 further showed that non-negligible disparate impacts, including both "the rich get richer" and "the poor get poorer", exist in the current SOTA SSL algorithms. Note Figure 2 and Figure 3 tested balanced datasets. Thus the resulting disparate impact would not be as severe as the unbalanced dataset.
> **Nonetheless, in the two sub-figures on the left panel of Figure 3(a), we do observe negative benefit ratios (up to -10\%) in the shaded area when the initial accuracy is around 10\%**. The dashed line indicates the mean of different trials and the shaded area indicates the standard deviation. Thus the negative shaded area shows the trials with up to -10\% (negative) benefit ratios. We also mentioned this observation on Page 8, above Section 5.2 that **"for some classes in CIFAR-100, this ratio can even go negative."** Besides, in our initial submission, we used red color to highlight the negative benefit ratios in Table 1. For example, the sub-population $t\_{white}$ (white identity in toxic labels) has a **negative benefit ratio (-9.57\%) in the setting of "Unbalanced (400)"**. All the negative benefit ratios mentioned above indicate "the poor get poorer."
>
> The phenomenon that several poor classes also benefit from SSL is mainly due to two reasons:
> * In a resource-constrained scenario, "the rich get richer" is likely to happen with "the poor get poorer". However, benefiting from the large capacity of deep neural networks and powerful SSL algorithms, the available "resource" after SSL is usually greater than before SSL. Thus the poor may not always get poorer.
> * MixMatch is a very powerful SSL algorithm. It is reasonable to believe the re-labeling procedures in each epoch in MixMatch could reduce the disparate impact compared to the two-iteration scenario considered in our theoretical analysis. This may also explain why "some poor classes are benefited."

---

> > ### Comment · Reviewer_x3NJ · 2021-11-20
> > **Further Discussion on Q1**
> >
> > Thanks for the detailed response. I am only concerned of my first three points, please take care of the minor concerns in your draft.
> >
> > Your response to Q1 is confusing to me, you start by saying that you are only interested in the "rich getting richer" part, and then go on to provide more evidence to why poor can get poorer in some cases.
> > In Figure 3(a), where you claim negative benefit ratio, I only see the left-most part of the left figure where benefit ratio is negative with no statistical significance. Also, for the dog case of Figure 1. So, I am still not convinced of the poor getting poorer part.
> >
> > Regarding the second concern, how important is poor getting poorer part of the Matthew effect for your fairness argument? Your response to my second concern relies on poor getting poorer part again, which I am yet not convinced about.
> >
> > I am more interested in finding out why some poor could get poorer instead of why some poor could get richer. Some poor groups benefiting from SSL is not that surprising and as far as I see, your analysis already can explain that.

---

> > > ### Author Response · Authors · 2021-11-21
> > > **Further Discussion [part 1]**
> > >
> > > Thanks for your follow-up questions, which indeed help us better present this paper. Your concerns are addressed as follows.
> > >
> > > **Question 1:**
> > > Your response to Q1 is confusing to me, you start by saying that you are only interested in the "rich getting richer" part, and then go on to provide more evidence to why poor can get poorer in some cases. In Figure 3(a), where you claim negative benefit ratio, I only see the left-most part of the left figure where benefit ratio is negative with no statistical significance. Also, for the dog case of Figure 1. So, I am still not convinced of the poor getting poorer part.
> > >
> > > **Response 1:**
> > >
> > > Sorry for the confusion. Response 1 (and our paper) mainly intends to show 1) "the rich getting richer" is commonly noted; 2) although we cannot consistently observe "the poor getting poorer", there is a possibility that it happens. Our observation is not exactly the same as the typical Matthew effect implies where the rich getting richer always happens with the poor getting poorer. To be more rigorous, we only highlight the "rich getting richer" part in our title. But **we are interested in both cases and think they both have fairness concerns**: of course, when "poor getting poorer" happens, this is probably a more obvious fairness issue. But we want to argue that even with only "rich getting richer", our results reveal substantial disparities in the benefits: some groups only benefit marginally from existing SSL algorithms (small benefit ratio 10\%) while the others more substantially (close to 80\% for CIFAR-100, Figure 3(a)). The lower-improved groups might still end up with a model that is less satisfactory when deployed in practice, while the richer groups can safely rely on a quality model. We view this disparity to be a potential fairness concern to be discussed by the ML and SSL community.
> > >
> > > **Figure 1 and Figure 3(a):** Yes, we agree that the "poor getting richer" case in both figures is not statistically significant, but it possibly happens (as in the above response). Note experiments in Figure 3(a) are run 5 times for stability, where the dashed line is the mean and the shaded area illustrates the standard deviation. It is worth noting that, all the results are plotted when the model is converged. Thus **the uncertainty in the shaded area is caused by different possible local optima, rather than the statistical error in estimating accuracies or benefit ratios.** Therefore, for example, in Figure 3(a), a benefit ratio of -10\% when the baseline accuracy is 10\% shows that, for this particular sub-population, **with a non-negligible probability**, the SSL model will converge to a model that is **10\% worse than the baseline**, which consolidates the "poor getting poorer" part. We revised the caption of Figure 3 to highlight the meaning of the shaded area.
> > >
> > > In **Figure 1(b)**, we first want to note the y-axis is not the benefit ratio, but the test accuracy. So when reading this figure, one shouldn't compare the numbers to 0 when looking for ``poor getting poorer" cases, but rather to compare the obtained accuracy from later epochs after using SSL with the accuracy at epoch 0, when only the supervised baseline was used. The class dog in **Figure 1(b)** shows the possibility of **"poor getting poorer" in an unbalanced case** where the supervised instances of dog are relatively less: at epoch zero (without SSL, supervised baseline), the accuracy for the dog class is **5\%**. But at epoch $10(\times 25)$ after using SSL, the accuracy dropped to close to **0\%**. The accuracy slightly bounced back at late epochs but remains lower than the initial accuracy. Note given the same supervised and unsupervised datasets, the "poor getting poorer" shown in class dog of Figure 1(b) is stable and robust to the change of random model initialization.
> > >
> > > We would also like to invite the reviewer to further check the first row of Table 1, which directly implements SSL on an originally unbalanced dataset. In the first row, **3 of 8 sub-populations have negative benefit ratios**, revealing severe fairness concerns.
> > >
> > > In summary, there are **two important messages** we want to present: **1) "rich getting richer" is common; 2) "poor getting poorer" may happen.** We clarified the two important messages in Motivating Example of Introduction during revision.

---

> > > > ### Comment · Reviewer_x3NJ · 2021-11-29
> > > > **Thanks for the response**
> > > >
> > > > "poor getting poorer may happen", but is quite rare in the experiments of the paper. I suggest the authors tone down the "poor getting poorer" in writing as well and carefully qualify "may happen" part to avoid reader confusion.
> > > >
> > > > The rich getting richer part in itself can raise a fairness concern since the model after SSL can fare even worse wrt equality of opportunity like fairness metric. Although, SSL causing "rich getting richer" effect is not so surprising, it is of practical fairness concern nevertheless.
> > > >
> > > > I raised my score a bit.
> > > >
> > > > Thanks.

---

> > > > > ### Author Response · Authors · 2021-11-29
> > > > > **Thank you for your comments!**
> > > > >
> > > > > Dear Reviewer x3NJ,
> > > > >
> > > > > Thank you for the active discussion, valuable comments, and supporting our paper. We will tone down the "poor getting poorer" part in writing and add more discussions to explain why it only "may happen". We will also polish our paper to better present the "rich getting richer" part.
> > > > >
> > > > > Best,
> > > > >
> > > > > ICLR 2022 Conference Paper469 Authors

---

> > > ### Author Response · Authors · 2021-11-21
> > > **Further Discussion [part 2]**
> > >
> > > **Question 2:**
> > > Regarding the second concern, how important is poor getting poorer part of the Matthew effect for your fairness argument? Your response to my second concern relies on poor getting poorer part again, which I am yet not convinced about.
> > >
> > > **Response 2:**
> > >
> > > We treat the **"poor getting poorer" as a severe and practical fairness concern (but not guaranteed to be observable in every setting)**, so we emphasize the possibility of "poor getting poorer" in our second response. In addition to "poor getting poorer," "rich getting richer" is more common. For example, in Figure 3(a), the common observation is that most of the sub-populations are benefited from SSL, but some of them obtain only very limited benefits. Take the third subplot of Figure 3(a) for an example, the sub-population whose baseline accuracy is less than 30\% tends to get a benefit ratio of less than 50\%, while the "richer" groups tend to reach 80\% of a benefit. Note MixMatch is demonstrated to be very powerful on CIFAR thus getting a benefit ratio of less than 50\% means an unsatisfying result. As discussed at the end of Response 1 in Further Discussion [part 1], we believe this significantly different benefit is also a fairness concern and calls for attention to consider improving the "poorer" groups when developing SSL algorithms.
> > >
> > > **Question 3:**
> > > I am more interested in finding out why some poor could get poorer instead of why some poor could get richer. Some poor groups benefiting from SSL is not that surprising and as far as I see, your analysis already can explain that.
> > >
> > > **Response 3:**
> > >
> > > The high-level intuition of "poor getting poorer" is that low baseline model performance leads to low accuracy in pseudo-labels. Applying SSL will induce the model to **memorize the wrong pseudo-labels, leading to a performance drop**. This intuition was shown **at the beginning of Section 3 (paragraph Intuition)** in our initial submission.
> > >
> > > We also have the corresponding theoretical analyses. Particularly, we first get **worst-case upper bounds for the baseline (Theorem 1) and SSL (Theorem 2)**. Particularly, the main factors are baseline model performance $\eta$ and the number of effective i.i.d. instances. Then by comparing both upper bounds, we present a **benefit ratio proxy in Corollary 1**. Below Corollary 1, we analyze the reason why we have "poor getting poorer" (copied from the initial submission): "SSL may have a negative effect on sub-population $\mathcal P$ if $\frac{\eta }{\Delta(N_{\mathcal P},N_{\mathcal{P}_L})} > 1$, i.e., **the benefit from getting more effective *i.i.d.* instances is less than the harm from wrong pseudo-labels.**" This negative effect indicates ”the poor get poorer.” According to your comments, we also revised Takeaways at the bottom of Page 6 and the paragraph below Corollary 1.
> > >
> > > -----------------------
> > >
> > > Thank you again for the fast response and valuable comments! Please feel free to let us know if anything needs further clarification.

---

> ### Author Response · Authors · 2021-11-18
> **Response to Reviewer x3NJ (part 2)**
>
> **Question 2: What is the fairness concern here?** SSL training benefit some/rich sub-population more than the others. As a result, it improves the average accuracy without hurting the worst group accuracy (since BR$\ge$0 for all groups). I would not call out SSL as unfair unless the poor population gets even worser, i.e. the rich groups get richer at the expense of poor groups. The authors should be more formal when discussing the fairness aspect and explain depreciating population disparity in the context of a fairness metric.
>
> **Response 2:**
>
> Figure 2 and Figure 3 mainly show that the disparate impact exists even in balanced datasets with SOTA SSL algorithms. As mentioned in the previous response, we observe **negative ratios (up to -10\%)** in the left two sub-figures of Figure 3(a).
> On one hand, it may not be that unfair if the worst group is not hurt. But it cannot be fair since the worst group get almost no benefits after consuming much computation effort. On the other hand, for the generally unbalanced dataset (e.g., Jigsaw is naturally unbalanced), Table 1 shows several groups are hurt (highlighted in red colors). For example, the sub-population $t\_{white}$, $nt\_{black}$, and $nt\_{latin}$ have **negative benefit ratios (-9.57\%, -8.33\%, and -13.82\%)** in the setting of "Unbalanced (400)". Kindly remind that, as mentioned in our initial submission (above Section 5.2): the disparate impact on the demographic groups in Jigsaw raises fairness concerns in real-world applications.
>
>
> **Question 3: Analysis.** Relationship between $\bar \eta'(\hat f_{D_L})$, $R_D(\hat f_{D_L})$, and $R_D(\hat f_{D_L\cup D_U})$ and their worst-case upper bounds.
>
> **Response 3:**
>
> Thanks for the interesting observation.
> Yes, in the confident case of Lemma 2, $\bar \eta'(\hat f_{D_L})$ is the expected error rate on distribution $\mathcal D$, i.e., $R_{\mathcal D}(\hat f_{D_L})$.
> But it is worth noting that, the upper bound is a worst-case bound, while $\eta$ captures the expected risk (a particular case). **Generally, it may not be fair to say which one is better by comparing a particular $R_{\text{supervised}}$ to the worst-case upper bound of $R_{\text{SSL}}$.**
> One counter-example is that, the converged local optima of supervised learning and SSL can be viewed as independent random variables (randomness due to different trials). Thus we cannot simply think that one particular supervised error is the same as the corresponding part in the worst-case bound of SSL.
> A relatively fair comparison is to check the upper bounds for both cases. Particularly, as shown in Theorem 1, given a small size of labeled dataset $D_L$, the supervised error is upper-bounded by
> \\\[
> R_{\text{supervised}} = R_{\mathcal D}(\hat f_{D_L}) \le \sqrt{\frac{2\log(2/\delta)}{N_L}} + \mathbb P(Y^*\ne Y).
> \\\]
>
> According to Theorem 2, the SSL error is upper-bounded by
> \\\[
> R_{\text{SSL}} = R_{\mathcal D}(\hat f_{D_L \cup D_U}) \le \bar \eta(\hat f_{D_L}) +  \sqrt{\frac{2\log(2/\delta)}{N}} + \mathbb P(\widetilde Y^*\ne \widetilde Y).
> \\\]
> This bound shows that, SSL reduces the data dependency error from $O(\sqrt{1/N_L})$ to $O(\sqrt{1/N})$ at the cost of introducing bias $\bar \eta(\hat f_{D_L})$. Note $N_L \ll N$ in general. In the **Wrap-up** paragraph, we show the bounds for Term-1 ($\eta$ term) are tight enough; thus, the above two bounds have similar tightness. It is reasonable to compare them and show some intuitions. In contrast, if we compare expectation of $R_{\text{supervised}}$ and the upper bound of $R_{\text{SSL}}$, we may get trivial results since they characterize different dimensions of the performance (one on average, one worst-case high probability bound).
>
>
> We also want to highlight the possibility of SSL performing worse or equally bad as the supervised baseline.
> In particular, if the model after SSL perfectly overfits the noisy distribution, it may not generalize better than the model before SSL (your inequality is true in the worst case). To better understand this, we can consider the following example. The model learned from the small labeled dataset results in $20\%$ of the error rate in the unsupervised dataset and $N_U\rightarrow \infty$. The unsupervised dataset is sufficiently large such that it covers all the test cases. In this case, fully memorizing the noisy labels of the unsupervised dataset has $20\%$ expected generalization error, which is the same as the model before SSL. Therefore, the special case you are considering exists, but it does not indicate that our bound is useless.

---

> ### Author Response · Authors · 2021-11-18
> **Response to Reviewer x3NJ (part 3)**
>
> **Question 4:** More technical details needed - Section 5.2
>
> **Response 4:**
>
> Thanks for the comments. Different settings are explained as follows.
> * Unbalanced (400): i.i.d. sampling $400$ labeled instances from the raw Jigsaw dataset.
> * Balanced labeled (400): For $400$ labels instances in "Unbalanced (400)", use a balanced sampling strategy during training to make sure each sub-population has equal probability (weights) in one epoch.
> * Balanced both (400): Like the previous case, but balance both labeled and unlabeled datasets.
> * Balanced both (800): Collect another 400 i.i.d. instances (800 in total) from the Jigsaw dataset and balance them
>
> Hope our explanations help you better understand this paper. We revised our paper based on your suggestions.
>
> **Question 5:**  Any BR value greater than 1 is only due to the empirical error in accuracy estimation?
>
> **Response 5:**
>
> Good point. Theoretically, your understanding may be one of the reasons for getting BR$>1$. In addition to your hypothesis, in practice, due to the complicated training procedures, sophisticated data augmentations, and the large capacity of deep neural networks, some sub-populations may perform even better than the ideal accuracy of training with standard supervised learning and standard data augmentations.
>
>
> **Question 6:** In the last line of A.3, Hoeffding's ineq. does not give uniform deviation bound unlike what was stated there.
>
> **Response 6:**
>
> Thanks for the comments. We've polished the proof, added more details, and fixed one typo ($2N \rightarrow N$). Please check the blue part in A.3 for more details.
>
>
> **Question 7:** The upper bound of Thm 1 contains constants that go with the $\sqrt{1/n}$ term, such as the maximum possible value of loss. But the order remains the same.
>
> **Response 7:**
>
> Our old Theorem 1 is established a general clean dataset $D$. To make it better connects to Theorem 2 and avoid confusion, we now re-state Theorem 1 on a small labeled dataset $D_L$. Then the bound is $R_{{\mathcal D}}(\hat f_{{D_L}})
> \le
>  \sqrt{\frac{2\log(2/\delta)}{{N_L}}} + \mathbb P(Y^*\ne Y).$
>  Note that $N_L \ll N$ in general.
>
>
> **Question 8:** In Tab.3 of appendix B.4, need to explain what 1:2 Benefit ratio, 1:5 accuracy etc. mean.
>
> **Response 8:**
>
> Thanks for pointing out the confusing places. It is actually "the benefit ratio of the case: CIFAR-100 1:2" and "the accuracy of the case: CIFAR-100 1:5", where CIFAR-100 1:2 means 40 instances per class for the first 50 fine classes, 20 instances per class for the next 50 fine classes, as mentioned in the caption of Table 3. Besides, Jigsaw (1.4:4.5:1:7.5) means that the ratios of different sub-populations' sizes are 1.4:4.5:1:7.5. We clarified this in Appendix C.4.

---

### Official Review · Reviewer_fRGJ · 2021-11-02

**Correctness:** 3
**Technical Novelty And Significance:** 4
**Empirical Novelty And Significance:** 4
**Recommendation:** 6
**Confidence:** 4

**Main Review:**

This paper aims to theoretically study the 'Matthew effect' (or disparate impact) of semi-supervised learning (SSL). The authors consider the consistency regularization-based approaches with either explicit or implicit pseudo labels. The authors first show that both types could be unified into one framework that is related to empirical risk minimization, where the pseudo labels are served as the noisy labels. Then the source of the Matthew effect is studied by deriving the generalization bound of the expected risk. A new metric named benefit ratio is further proposed to theoretically understand the Matthew effect of SSL with consideration of the derived generalization bounds. Experimental results shows that (1) popular SSL methods almost all suffer from this effect, and (2) the mitigation strategies guided by analyzing benefit ratio could reduce such Matthew effect.

Overall, I think this paper is studying a very interesting and important problem. Theoretically analyzing the varied performance over different sub-populations may help design a better SSL method so that certain sub-populations are taken care of, which might also improve the overall performance. Please see my concerns regarding the submission as follows.

Concerns
- Usually in algorithmic fairness, disparate impact is more concerned about the acceptance rate rather than accuracy (though it may not be correct in legal support area). I think it would be better to have a few sentences to explicitly clarify the difference between the 'disparate impact' in this paper and the 'disparate impact' usually used in algorithmic fairness.

- Some key references are missing, which study the varied classification accuracy among different sub-populations. Please see these papers at the end of my comment. And I believe the authors should discuss whether the existing works can be used in SSL setting. If yes, some analysis on their performance are needed; if not, some sentences on discussing why they cannot work are needed. My hunch is that, if we can view SSL as supervised learning with noisy labels, some of these works should work.

- In RHS of Eq. (2), do we really need the average over the expectation? I thought the average should be absorbed into the expectation. I hope the authors could clarify it.

- In proof of Lemma 3, the denominator in the square root has a coefficient 2 while the results in the main body does not have this coefficient, so at least one place is wrong.

- Again, in proof of Lemma 3, is it possible for the authors to provide more details on how the first inequality (<= 0 + 2max|...|) is derived?

- In figure 2, the correlation between baseline accuracy and benefit ratio is not clear for UDA on Yahoo dataset. Some analysis are needed. Also, the authors might want to correct their claim 'We consistently observe that the class labels with higher baseline accuracies have higher benefit ratios on both CIFAR-10 and Yahoo! Answers datasets.' because results of UDA on Yahoo does not reveal such observation.

- Just out of my curiosity (this may be out of the scope of this paper), I wonder whether similar analysis can be generalized to SSL on graphs.

References
* Hashimoto, T., Srivastava, M., Namkoong, H., & Liang, P. (2018, July). Fairness without demographics in repeated loss minimization. In International Conference on Machine Learning (pp. 1929-1938). PMLR.
* Kim, M. P., Ghorbani, A., & Zou, J. (2019, January). Multiaccuracy: Black-box post-processing for fairness in classification. In Proceedings of the 2019 AAAI/ACM Conference on AI, Ethics, and Society (pp. 247-254).
* Lahoti, P., Beutel, A., Chen, J., Lee, K., Prost, F., Thain, N., ... & Chi, E. H. (2020). Fairness without demographics through adversarially reweighted learning. arXiv preprint arXiv:2006.13114.
* Diana, E., Gill, W., Kearns, M., Kenthapadi, K., Roth, A., & Sharifi-Malvajerdi, S. (2021). Multiaccurate Proxies for Downstream Fairness. arXiv preprint arXiv:2107.04423.
* Martinez, N., Bertran, M., & Sapiro, G. (2020, November). Minimax pareto fairness: A multi objective perspective. In International Conference on Machine Learning (pp. 6755-6764). PMLR.

**Summary Of The Paper:**

Please see main review.

**Summary Of The Review:**

This paper studies an interesting problem. But some key references are missing, which should be discussed in related work and (potentially) evaluated as baseline methods.

---

> ### Author Response · Authors · 2021-11-18
> **Response to Reviewer fRGJ (part 1)**
>
> Thanks for supporting our paper's technical and empirical novelty and providing references to help us justify our work. We have cited all the references and discussed them in Related Works and other necessary places. Your concerns are addressed as follows.
>
> **Question 1:** Usually in algorithmic fairness, disparate impact is more concerned about the acceptance rate rather than accuracy (though it may not be correct in legal support area). I think it would be better to have a few sentences to explicitly clarify the difference between the 'disparate impact' in this paper and the 'disparate impact' usually used in algorithmic fairness.
>
> **Response 1:**
>
> Our investigation of disparate impact is similar to the Equal Odds [R1] definition that captures the differences in achieved model accuracy. More specifically, reducing the disparate impact in SSL means pursuing an "Equal Benefit Ratio" solution. The accuracy for each sub-population follows the same intuition as "multiaccuracy" defined in your provided references. We have clarified this in the first and third paragraphs of the Introduction (highlighted in blue).
>
> [R1] Hardt, M., Price, E. and Srebro, N., 2016. Equality of opportunity in supervised learning. Advances in neural information processing systems, 29, pp.3315-3323.
>
>
> **Question 2:** Some key references are missing, which study the varied classification accuracy among different sub-populations. Please see these papers at the end of my comment. And I believe the authors should discuss whether the existing works can be used in SSL setting. If yes, some analysis on their performance are needed; if not, some sentences on discussing why they cannot work are needed. My hunch is that, if we can view SSL as supervised learning with noisy labels, some of these works should work.
>
>
> **Response 2:**
>
> Thanks for providing more related works. We attempt to extend the concept of "disparate impact" to SSL, which should capture the change of accuracies before and after SSL. This motivates us to define the benefit ratio in Section 4. To the best of our knowledge, existing works mainly focus on fairness in supervised learning, but supervised learning, noisy learning, semi-supervised learning are three different topics. We acknowledge there are some relations among the three topics, but the definitions and analyses on their fairness issues are quite different. For example, the provided references mainly focus on fairness in supervised learning, which is different from the fairness in noisy learning since the noise rate of different groups may cause fairness issues [R2]. In addition, [R2] is built on a robust loss function, peer loss, and studies the particular group-dependent label noise. In our analyses, we theoretically analyze the disparate impact in SSL with the help of pseudo-labels. We acknowledge pseudo-labels have some connections to noisy labels, but our targeted methods (MixMatch, UDA, and MixText) did not apply any robust loss function. Moreover, the label noise in SSL cannot be simply modeled as group-dependent. We appreciate your careful thinking about related works. Hope our clarifications could better highlight our novelty.
>
> In our revision, we cited these references and clarified differences in the paragraph *Disparate impact* included in Related Works.
>
> [R2] Wang, J., Liu, Y., \& Levy, C. (2021, March). Fair classification with group-dependent label noise. In Proceedings of the 2021 ACM Conference on Fairness, Accountability, and Transparency (pp. 526-536).
>
> **Question 3:** In RHS of Eq. (2), do we really need the average over the expectation? I thought the average should be absorbed into the expectation. I hope the authors could clarify it.
>
>
> **Response 3:**
>
> Thanks for your detailed technical question. Kindly remind that the expectation is taken only over noisy label $\widetilde Y$. The average over $n$ is still the ERM. By this transformation, the soft label $\tilde {\bf{y}}$ turns to a hard label $\widetilde Y$ that takes different values with different probabilities. We use this transformation to facilitate our derivations.

---

> ### Author Response · Authors · 2021-11-18
> **Response to Reviewer fRGJ (part 2)**
>
> **Question 4:** In proof of Lemma 3, the denominator in the square root has a coefficient 2 while the results in the main body does not have this coefficient, so at least one place is wrong. Again, in proof of Lemma 3, is it possible for the authors to provide more details on how the first inequality ($\leq$ 0 + 2max$|$...$|$) is derived?
>
> **Response 4:**
>
> We really appreciate your detailed review of this paper. The denominator in Lemma 3 should be $\sqrt{N}$. We've fixed the typo in the revised version. We also added more details and highlighted them in blue to explain how the first inequality is derived.
> Particularly, this inequality holds since:
> * $\hat R_{D}(\hat f_{D})$ achieves the minimum empirical risk according to its definition, thus $\hat R_{D}(\hat f_{D}) - \hat R_{D}(f^*_{\mathcal D})\le 0$;
> * each of the following two terms will be no greater than the corresponding maximal gap $\max_{f\in\mathcal F} |\hat R_{D}(f) - R_{\mathcal D}(f)|$.
>
> **Question 5:** In figure 2, the correlation between baseline accuracy and benefit ratio is not clear for UDA on Yahoo dataset. Some analysis are needed. Also, the authors might want to correct their claim `We consistently observe that the class labels with higher baseline accuracies have higher benefit ratios on both CIFAR-10 and Yahoo! Answers datasets.' because results of UDA on Yahoo does not reveal such observation.
>
> **Response 5:**
>
> Thanks for your comment. We agree the variance is large in this case, but the dashed line from linear regression also shows the benefit ratio is increasing with baseline accuracy as the other subfigures do. This variance may be caused by deep neural networks and the stability of UDA itself. Besides, we changed "consistently" to "statistically" in our revised version based on your comment. Thanks again!
>
> **Question 6:** Just out of my curiosity (this may be out of the scope of this paper), I wonder whether similar analysis can be generalized to SSL on graphs.
>
> **Response 6:**
>
> Thanks for the interesting question. Our Theorem 2 shows that the SSL with consistency regularization provides pseudo labels, which is the main cause of disparate predictions. So we hypothesize that the observations would hold for SSL on graphs if consistency regularization is adopted and pseudo labels are generated. But it is possible that the graph structure can help improve the quality of the pseudo labels, therefore reducing the observed disparity.

---

> > ### Comment · Reviewer_fRGJ · 2021-11-21
> > **Response to authors**
> >
> > Thank you for taking the effort to address my concerns. Most of my concerns are addressed, so I will raise my score later. In terms of Reviewer x3NJ's concern on 'poor getting poorer', I am ok with the current submission, but I do think x3NJ raises some concern that could mislead readers. I think the authors should be more careful with the tone when describing or discussing about 'poor getting poorer'.

---

> > > ### Author Response · Authors · 2021-11-22
> > > **Thank you for your valuable comments!**
> > >
> > > Thanks a lot for all these detailed and valuable comments and references. We would also like to thank you for the fast response and for supporting our paper. We are glad that most of your concerns are addressed. We will make the presentation clearer and better describe the "poor getting poorer" part to avoid confusion and highlight our contribution.

---

### Official Review · Reviewer_FRon · 2021-11-02

**Correctness:** 3
**Technical Novelty And Significance:** 3
**Empirical Novelty And Significance:** 3
**Recommendation:** 6
**Confidence:** 4

**Main Review:**


Strengths:

- The paper provides meaningful analyses to investigate the disproportionate performance of semi-supervised learning (SSL) on different subgroups.
- The theoretical results show that the SSL frameworks possibly provide discriminative benefits to the subpopulations, which is consistent with the intuition.
- In addition, experiments are performed with three representative SSL methods on several datasets for vision and natural language processing tasks. The experimental results confirm that the discrimination of the SSL frameworks can be a problem in various datasets.

Concerns:

- It would be better to clarify the term “disparate impact.” Disparate impact is used in many model fairness papers as a specific category of unfairness or a group fairness metric [1, 2]. Thus, if the paper used “disparate impact” as a broader term to explain the bias in SSL, it would be nice to explain the difference from the language in the previous papers.

- In Figure 1, several “poor” classes seem to get enough accuracy gain. The intuition for “the rich get richer, and the poor get poorer” is convinced, but not sure why Figure 1 can be the best example for this message. For example, “bird, cat, and deer classes” in the left subfigure and “horse class” in the right subfigure initially have poor accuracy, but the performance adequately increases during SSL. Thus, I am not sure this figure clearly shows the main message. It would be nice if the paper clarifies these phenomena and connect them to the main takeaway.

- It would be helpful if the paper gives more details on Section 3. For example, $\eta$ is considered the disparity due to baseline and the major source of disparity, but it is not clearly explained. Also, the warm-up paragraph says the range of Term-1 becomes [1-2$\epsilon$, 1]. However, if the range changes to the corresponding bound, $\eta$ and $e$ should become 1, which is unclear. Such details might help to strengthen the paper.

- When evaluating the training by the benefit ratio, how can we get the ideal accuracy (i.e., the first term of the denominator)? Since the ideal accuracy is defined as the accuracy from fully-supervised learning with the ground truth labeled dataset, it seems hard to get the value in the real scenarios. Also, is there any reason why the existing group fairness metrics (e.g., demographic parity and equalized odds) cannot be applied in the SSL scenario? These might provide other intuitions on the SSL frameworks.

- Some experimental settings are unclear. For example, why are the experiments assumed that the labeled data are balanced, but the unlabeled data are imbalanced  (Figures 2 & 3)? It would be better to explain why the current setting is a suitable and natural choice for observing fairness in SSL.

References

[1] Zafar et al., Fairness Constraints: Mechanisms for Fair Classification, AISTATS 2017.

[2] Bellamy et al., AI Fairness 360: An Extensible Toolkit for Detecting, Understanding, and Mitigating Unwanted Algorithmic Bias, IBM J. Res. Dev. 2019.



**Summary Of The Paper:**

This paper analyzes the discriminative performances of semi-supervised learning (SSL) on different subgroups, based on the key message that the already accurate group gets more benefits from the SSL. The paper first proposes a unified loss for analyzing the SSL frameworks. Then, the theoretical analyses show how the SSL produces discriminative performances on the different groups. In addition, a metric called benefit ratio is suggested to measure how much the learning benefits a certain subgroup. Experiments show overall consistent results with the main message from the theory.


**Summary Of The Review:**

This paper provides useful analyses on the fairness issue in the semi-supervised setting. Although I have some comments as described above, the paper is overall well-written.
Thus, I believe that this paper is worth publishing after clarifying several parts.

---

> ### Author Response · Authors · 2021-11-18
> **Response to Reviewer FRon (part 1)**
>
> Thank you for supporting our paper and providing valuable comments. We've polished our paper based on your comments. Your concerns are addressed as follows.
>
> **Question 1:** It would be better to clarify the term “disparate impact.” Disparate impact is used in many model fairness papers as a specific category of unfairness or a group fairness metric [1, 2]. Thus, if the paper used “disparate impact” as a broader term to explain the bias in SSL, it would be nice to explain the difference from the language in the previous papers.
>
> **Response 1:**
>
> Thanks for supporting our paper and providing more related works. Yes, we attempt to extend the concept of ``disparate impact'' to SSL, which should capture the change of accuracies before and after SSL. This motivates us to define the benefit ratio in Section 4. We cited the references and clarified them at the bottom of Page 1 (highlighted in blue). We also discussed the relations between our disparate impact and the existing fairness metrics in the first paragraph of the Introduction and the paragraph below Figure 1 (highlighted in blue). We will add more comprehensive discussions in our future revisions.
>
> **Question 2:** In Figure 1, several “poor” classes seem to get enough accuracy gain. The intuition for “the rich get richer, and the poor get poorer” is convinced, but not sure why Figure 1 can be the best example for this message. For example, “bird, cat, and deer classes” in the left subfigure and “horse class” in the right subfigure initially have poor accuracy, but the performance adequately increases during SSL. Thus, I am not sure this figure clearly shows the main message.
>
> **Response 2:**
>
> Good point. We do observe some "poor'' classes may benefit from SSL. But it is also worth noting that, as mentioned in Motivating Example (shown in Introduction), some "poor'' sub-population, such as dog that has a low baseline accuracy, will remain a low-level performance in Figure 1(a) or even **get worse performance in Figure 1(b)**. Particularly, Figure 1(b) shows that class dog has an initial test accuracy of about $5\%$ at epoch 0 but it **drops** to almost $0\%$ accuracy at epoch 250. This observation supports the claim "the poor get poorer''.
>
> Our Figure 2 and Figure 3 further showed that non-negligible disparate impacts, including both "the rich get richer'' and "the poor get poorer'', exist in the current SOTA SSL algorithms. For instance, in the two sub-figures on the left panel of Figure 3(a), we observe **negative benefit ratios (up to -10\%)** in the shaded area when the initial accuracy is around 10\%. Note the dashed line indicates the mean of different trials and the shaded area indicates the standard deviation. Thus the negative shaded area shows the trials with up to -10\% (negative) benefit ratios. Besides, in our initial submission, we used red to highlight the negative benefit ratios in Table 1. For example, the sub-population $t\_{white}$ (white identity with toxic labels) has a **negative benefit ratio (-9.57\%) in the setting of "Unbalanced (400)''**. All the negative benefit ratios mentioned above indicate "the poor get poorer''.
>
> We'd also like to point out that we only briefly mentioned the possibility of having the Matthew effect. We are aware that, in most cases, we only observe "the rich get richer'', which is the only direction that is emphasized in the title.
>
> **Question 3:** It would be nice if the paper clarifies these phenomena (several poor classes get enough accuracy gain) and connect them to the main takeaway.
>
> **Response 3:**
>
> Thanks for your suggestion. The phenomenon that several poor classes also benefit from SSL is mainly due to two reasons:
>
> * In a resource-constrained scenario, "the rich get richer'' is likely to happen with "the poor get poorer''. However, benefiting from the large capacity of deep neural networks and powerful SSL algorithms, the available "resource'' after SSL is usually greater than before SSL. Thus the poor may not always get poorer.
> * MixMatch is a very powerful SSL algorithm. It is reasonable to believe the re-labeling procedures in each epoch in MixMatch could reduce the disparate impact compared to the two-iteration scenario considered in our theoretical analysis. This may also explain why "some poor classes are adequately benefited.''
>
> We added more discussions in Appendix B in our revision.

---

> > ### Comment · Reviewer_FRon · 2021-11-22
> > **After reading the authors' response**
> >
> > Thanks for the careful answers. Most of my concerns are now addressed.
> > However, as the authors mentioned in their response, a more detailed discussion on the term "disparate impact" might be needed to strengthen the paper's clarity.

---

> > > ### Author Response · Authors · 2021-11-23
> > > **Thank you**
> > >
> > > Thank you for supporting our paper and providing these insightful questions/comments. We are glad that most of your concerns are addressed. We will add more discussions on the term "disparate impact" in future versions.

---

> ### Author Response · Authors · 2021-11-18
> **Response to Reviewer FRon (part 2)**
>
> **Question 4:** It would be helpful if the paper gives more details on Section 3. For example,  is considered the disparity due to baseline and the major source of disparity, but it is not clearly explained. Also, the warm-up paragraph says the range of Term-1 becomes $[1-2\epsilon, 1]$. However, if the range changes to the corresponding bound,  and should become 1, which is unclear. Such details might help to strengthen the paper.
>
> **Response 4:**
>
> Thanks for the suggestion. $\eta$ and $e$ are defined in the paragraph before Lemma 1. We will better organize this section and highlight important information in our future revisions. In the wrap-up paragraph, when $Y|X$ is confident as Lemma 2, we have $\bar \eta = \bar e$. Additionally, when $\widetilde A_{\hat f_{\widetilde D}} = 1-\epsilon$, the range of Term-1 becomes $[(1-2\epsilon)\eta,\eta]$ (sorry for missing $\eta$ here and causing confusion). Note $\epsilon$ depends on how a classifier learns the noisy distribution, which is independent of $\eta$. Thus we did not require $\eta=1$ when the range changes to the corresponding bound. Thanks for the careful reviewing. Sorry for the confusion caused by missing $\eta$.
>
> **Question 5:**  When evaluating the training by the benefit ratio, how can we get the ideal accuracy (i.e., the first term of the denominator)? Since the ideal accuracy is defined as the accuracy from fully-supervised learning with the ground truth labeled dataset, it seems hard to get the value in the real scenarios. Also, is there any reason why the existing group fairness metrics (e.g., demographic parity and equalized odds) cannot be applied in the SSL scenario? These might provide other intuitions on the SSL frameworks.
>
> **Response 5:**
>
> Yes, calculating the ideal accuracy requires a fully-supervised dataset. Our proposed benefit ratio metric focuses on a posterior evaluation to justify whether an SSL algorithm is fair or not. It has the potential of providing guidance and intuitions for the design of fair SSL algorithms on standard datasets with full ground-truth labels. Whether a fair SSL algorithm on one dataset is still fair on another dataset would be interesting future work. In real scenarios without full supervision, we may use some extra knowledge to estimate the upper bound of the accuracy of each sub-population and set it as a proxy of the ideal accuracy.
>
>
> Our main motivation is capturing the disparate improvement of accuracies before and after SSL. Current fairness metrics focus more on the static scenario and do not address the dynamical changes of accuracies in each sub-population.
>
> **Question 6:** Some experimental settings are unclear. For example, why are the experiments assumed that the labeled data are balanced, but the unlabeled data are unbalanced (Figures 2 \& 3)? It would be better to explain why the current setting is a suitable and natural choice for observing fairness in SSL.
>
> **Response 6:**
>
> We would like to clarify that **both the labeled and the unlabeled data are balanced in Figures 2 \& 3**. In Section 5.1, we mainly show that, even though the size of each sub-population is equal, disparate impacts exist in the model accuracy of different sub-populations. Thus we require balanced datasets in Figures 2 and 3. Note the original CIFAR dataset and Yahoo! Answers dataset are balanced. We sampled a balanced subset from the raw (unbalanced) Jigsaw dataset. We also tested i.i.d. sampled (unbalanced) Jigsaw dataset in Table 1 (Unbalanced 400). We clarified the corresponding settings in Section 5.1 and Section 5.2 (highlighted in blue).

---

### Official Review · Reviewer_fzF9 · 2021-11-03

**Correctness:** 4
**Technical Novelty And Significance:** 3
**Empirical Novelty And Significance:** 3
**Recommendation:** 6
**Confidence:** 3

**Main Review:**

The paper studies the sources of disparate impact on SSL from both theoretical and experimental demonstrations. Theoretically, the SSL problem was transformed to supervised learning with noisy labels by assigning pseudo labels. Both theoretical and empirical work show that the major source of the disparity comes from the low supervised error.

The paper has high ethical value. The theoretical proof and experiments are sound and support the conclusion of the paper.  I have a few minor comments for clarification.
- Could you please clarify why figure3(a) shows a range of the benefit ratio, not the other figures?
- How is the ideal ground truth label a_ideal(P) calculated in the experiments? How robust is that, and how will it affect the calculation of BR?


**Summary Of The Paper:**

This paper presents the disparate impact on semi-supervised learning, where the sub-population that has a higher baseline accuracy tends to benefit more from SSL. The paper targets an important problem to ensure the fairness of machine learning algorithms. The authors also proposed a new metric benefit ratio to facilitate the evaluation of SSL. They also suggested two methods to mitigate the disparate, balancing the labeled data or collecting more labeled data.

**Summary Of The Review:**

In general, the paper is well written. The task is novel and well supported by theoretical proof and experimental evaluations.

---

> ### Author Response · Authors · 2021-11-18
> **Response to Reviewer fzF9**
>
> Thanks for supporting our paper and providing valuable comments. We've polished our paper based on your comments. Your concerns are addressed as follows.
>
> **Question 1: Could you please clarify why figure3(a) shows a range of the benefit ratio, not the other figures?**
>
> **Response 1:**
>
> Thanks for supporting our paper and helping us better present this paper. We note that the training on CIFAR-100 is not stable. We ran the experiments for $5$ times with different initialization and random seeds to make the results more meaningful. Thus we have $500$ points (much more than other experiments) in each figure. To better show the trends of these points, we use a Gaussian kernel and plot the mean (dashed line) and standard deviation (shaded area) in Figure 3(a).  In our revision, it is clarified in the caption of Figure 3.
>
>  **Question 2: How is the ideal ground truth label a\_ideal(P) calculated in the experiments? How robust is that, and how will it affect the calculation of BR?**
>
> **Response 2:**
>
> Good question. $a_{\text{ideal}(\mathcal P)}$ is calculated by supervised learning on the full dataset. For example, in CIFAR-10, we train a ResNet34 by minimizing the cross-entropy loss among 50k labeled training instances. It is quite robust in some experiments, such as CIFAR-10. But it is not that robust in CIFAR-100 since we train only with coarse labels (20 classes, each coarse class contains 5 sub-populations with fine labels) and test with fine labels (100 classes). To make it robust, we did experiments for $5$ times as mentioned in our first response. We also note that there might be some outliers when the accuracies are not robust, which motivates us to use a Gaussian kernel to plot a smooth trend.

---

### Author Response · Authors · 2021-11-18
**General response to all reviewers**

We thank all the reviewers for their detailed and helpful comments. Based on the comments, we revised our paper and highlighted major revisions in blue. We will respond in detail to each reviewer individually. Please feel free to let us know if there is still any confusion.

---

### Decision · Program_Chairs · 2022-01-20

**Decision:**

Accept (Poster)

**Comment:**

The paper analyses theoretically the 'Matthew effect' (disparate impact) in the setting
of the semi-supervised learning and its effect on fairness and performance.

All reviewers agree that the paper deals with a very interesting topic and important problem.
The paper discusses and presents a thorough and convincing analysis of the effect.
There were multiple concerns raised mainly around the lack of clarity at parts of the paper.
The authors did a very good job at resolving those and bringing their submission to a good standard.
In the rebuttal I was glad to see a great dialog evolving among the authors and reviewers.
I congraultate both sides.

Happy to recommend acceptance.